



# Assessment of a full-field initialised decadal climate prediction system with the CMIP6 version of EC-Earth

Roberto Bilbao[1], Simon Wild[1], Pablo Ortega[1], Juan Acosta-Navarro[1], Thomas Arsouze[1], Pierre-Antoine Bretonnière[1], Louis-Philippe Caron[1], Miguel Castrillo[1], Rubén Cruz-García[1], Ivana Cvijanovic[1], Francisco Javier Doblas-Reyes[1,2], Markus Donat[1], Emanuel Dutra[3], Pablo Echevarría[1], An-Chi Ho[1], Saskia Loosveldt-Tomas[1], Eduardo Moreno-Chamarro[1], Nuria Pérez-Zanon[1], Arthur Ramos[1], Yohan Ruprich-Robert[1], Valentina Sicardi[1], Etienne Tourigny[1], and Javier Vegas-Regidor[1]

[1]Barcelona Supercomputing Center, Jordi Girona 29, 08034, Barcelona, Spain
[2]Institució Catalana de Recerca i Estudis Avançats (ICREA), Passeig Lluis Companys 23, 08010, Barcelona, Spain
[3]Instituto Dom Luiz, Faculdade de Ciências, Universidade de Lisboa, Lisboa, Portugal

**Correspondence:** Roberto Bilbao (roberto.bilbao@bsc.es)

**Abstract.** In this paper we present and evaluate the skill of the EC-Earth3.3 decadal prediction system contributing to the Decadal Climate Prediction Project - Component A (DCPP-A). This prediction system is capable of skilfully simulating past global mean surface temperature variations at interannual and decadal forecast times as well as the local surface temperature in regions such as the Tropical Atlantic, the Indian Ocean and most of the continental areas, although most of the skill comes from the representation of the externally forced trends. A benefit of initialisation in the predictive skill is evident in some areas of the Tropical Pacific and North Atlantic Oceans in the first forecast years, an added value that gets mostly confined to the southeast Tropical Pacific and the eastern Subpolar North Atlantic at the longest forecast times (6-10 years). The central Subpolar North Atlantic shows poor predictive skill and a detrimental effect of the initialisation due to the occurrence of an initialisation shock, itself related to a collapse in Labrador Sea convection by the third forecast year that leads to a rapid weakening of the Atlantic Meridional Overturning Circulation (AMOC) and excessive local sea ice growth. The shutdown in Labrador Sea convection responds to a gradual increase in the local density stratification in the first years of the forecast, ultimately related to the different paces at which surface and subsurface temperature and salinity drift towards their preferred mean state. This transition happens rapidly in the surface and more slowly in the subsurface, where, by the tenth forecast year, the model is still far from the typical mean states in the corresponding ensemble of historical simulations with EC-Earth3. Our study thus highlights the importance of the Labrador Sea for initialisation, the relevance of reducing model bias by model tuning or, preferably, model improvement when using full-field initialisation, and the need to identify optimal initialisation strategies.





# 1 Introduction

Interest in seasonal-to-decadal climate predictions has grown in recent years due to their potential to provide relevant cli-
mate information for decision making in different socio-economic sectors (e.g. Suckling, 2018; Solaraju-Murali et al., 2019;
Merryfield et al., 2020). Scientifically, climate predictions have provided novel ways of evaluating and comparing climate sim-
ulations with observations and improve our understanding of the intrinsic predictability of the climate system, including the
key mechanisms operating at interannual-to-decadal timescales.

On these time-scales a large part of the predictable signal of climate variations during the observational period is attributable
to changes in external radiative forcings (i.e. changes in the climate system energy balance), which can be of natural (e.g. solar
irradiance, volcanic aerosols) or anthropogenic origin (e.g. greenhouse gas concentrations, land use changes and anthropogenic
aerosols). For example, at the global scale most of the surface temperature changes can be explained by the warming trend
caused by the increasing atmospheric greenhouse gas concentrations, which is partly compensated by a parallel increase in
anthropogenic aerosols (e.g. Bindoff et al., 2013), and the sporadic cooling episodes that followed the major volcanic eruptions
of Mt Agung (1963), El Chichon (1982) and Pinatubo (1991) (e.g. Ménégoz et al., 2018; Hermanson et al., 2020). Estimates
of past changes in these radiative forcings are prescribed as boundary conditions to drive the so called historical climate
simulations, which investigate the influence of the forcings on the recent climate variations. These experiments are continued
into the future as climate projections with imposed anthropogenic radiative forcings that follow different theoretically derived
socio-economic emission scenarios (O'Neill et al., 2016).

The other main source of predictability originates from internal variability, in particular in the slowly evolving components of
the climate system (i.e. the ocean) (e.g. Meehl et al., 2009). Beside imposing external radiative forcings, climate predictions are
initialised from the observed state which phases internal variability with observations, taking advantage of its predictability.
Predictive skill of real-time forecast systems is assessed by producing retrospective predictions (or hindcasts), that are then
contrasted with observations. At seasonal to annual timescales, hindcasts show high levels of predictive skill for El Niño-
Southern Oscillation (ENSO) (e.g. Barnston et al., 2019). On decadal timescales, many climate models have also shown the
capacity to skillfully predict the Atlantic Multidecadal Variability (AMV) (e.g. García-Serrano et al., 2015), and to a lesser
extent the Interdecadal Pacific Oscillation (IPO) (e.g. Mochizuki et al., 2009; Chikamoto et al., 2015). The North Atlantic
Ocean, and more precisely the Subpolar Gyre, has been identified as a region where different retrospective prediction systems
skillfully predict the evolution of sea surface temperatures (SST) and upper ocean heat content (OHC) (e.g. Pohlmann et al.,
2009; Keenlyside et al., 2008; Robson et al., 2018; Yeager et al., 2018), although these same systems show limited capacity to
predict the climate of the neighboring continents, which might be related to an inaccurate representation of key teleconnection
mechanisms (e.g. Goddard et al., 2013; Doblas-Reyes et al., 2013). Encouragingly, a recent paper by Smith et al. (2020) has
shown that decadal predictions contributing to CMIP6 can be skillful at predicting the low-frequency variations of the North
Atlantic Oscillation (NAO), the leading mode of the winter atmospheric circulation in the Northern Hemisphere (Hurrell,
1996), when considering a large multi-model ensemble. The study concludes that the NAO and related climate signals over
Europe might be more predictable than models suggest, and that large ensembles of predictions are necessary to circumvent an





inherent problem of current forecast systems, the fact that they can strongly underestimate the predictable signals (Scaife and Smith, 2018).

To reinforce the inter-comparability of the results and allow for the exploitation of large multi-model ensembles, the decadal climate prediction community has promoted the development of coordinated climate prediction exercises. The Decadal Climate Prediction Project (DCPP; Boer et al., 2016), as part of the Coupled Climate Model Intercomparison Project Phase 6 (CMIP6; Eyring et al., 2016), and building upon CMIP5 (Taylor et al., 2012) and the efforts of previous projects (e.g. SPECS, ENSEMBLES), provides such a framework for addressing different aspects and current knowledge gaps of decadal climate prediction. DCPP-A is the main component and consists of an ensemble of decadal hindcasts, initialised at yearly intervals

from 1960 up to 2018 using prescribed CMIP6 external radiative forcings.

A crucial step to maximise the skill of decadal predictions is initialisation. It is of major importance to produce physically consistent initial conditions that reflect as faithfully as possible the observed state of climate. In particular, for the 3-dimensional ocean temperature and salinity fields, which determine the basin-wide density gradients and through them the large-scale ocean circulations and transports, which are essential for predictability on decadal timescales (e.g., Meehl et al.,

2014; Brune and Baehr, 2020). However, observational records are sparse in time and space, especially in the deep ocean, which posses a challenge to accurately constrain the initial state exclusively from observations. For this reason, a typical approach in climate prediction is to use initial conditions from ocean and atmosphere reanalysis. These are produced with data assimilation techniques that ensure a dynamically consistent estimation of the climate state that takes into account observational uncertainties.

Due to structural errors in climate models and biases in their climatologies, when initialised from the observed state, predictions suffer from initial shocks and drifts (e.g. Magnusson et al., 2012; Sanchez-Gomez et al., 2016; Kröger et al., 2018; Meehl et al., 2014). Initial shocks are abrupt changes that occur soon after initialisation as a result of the adjustment of the climate model to the initial state, while the drift represents the mean evolution of the forecasts towards the imperfect mean model climate. Two main approaches of initialisation are often used; 'full-field initialisation' which uses directly observational esti-

mates to initialise the model (e.g., Pohlmann et al., 2009), and 'anomaly initialisation' that imposes the observational estimate anomalies on the model climatology (e.g., Smith et al., 2007; Keenlyside et al., 2008). No clear advantage of one approach with respect to the other has been found in terms of forecast quality (e.g. Magnusson et al., 2012; Weber et al., 2015; Boer et al., 2016). The latter was specifically designed to reduce the forecast drift, as it implies initialising from a state closer to the model climatology. However, incompatibilities between the imposed anomalies and the typical model variability have been shown to

cause dynamical imbalances leading to biases in the predictions (e.g., Magnusson et al., 2012; Hazeleger et al., 2013; Volpi et al., 2017). The occurrence of forecast drifts and biases compromises the quality of the predictions, a problem that can be partly circumvented by correcting the predictions a posteriori, for example by computing forecast-time dependent anomalies (e.g., Meehl et al., 2014; Goddard et al., 2013; Choudhury et al., 2017).

With the objective of reducing initial shocks, several decadal forecast centres consider the production of in-house assimila-

tion experiments with the same model or model components used for the forecasts from which the initial states are derived. The simplest and commonly used assimilation framework consists in producing assimilation runs with individual model com-





ponents (referred to as 'weakly coupled'), typically of the ocean model, since it is the most important for the predictability on decadal timescales (e.g., Sanchez-Gomez et al., 2016; Servonnat et al., 2015). This method may benefit the initialisation of an individual model component, however initialisation shocks may occur due to incompatibilities among the initial conditions.

For this reason, many forecast centres are moving towards fully coupled assimilation (referred to as 'strongly coupled'), which is more technically challenging but assures physical consistency of the initial conditions of all the components, among other advantages (e.g., Brune and Baehr, 2020). For assimilation, a range of different techniques have been used to produce the reconstructions, from classical nudging approaches based on Newtonian relaxation (e.g., Sanchez-Gomez et al., 2016; Servonnat et al., 2015) to more complex and computationally expensive methods like the Ensemble Kalman Filter approach (e.g.,

Counillon et al., 2014; Dai et al., 2020), that take into account aspects of the observational uncertainty.

The aim of this paper is to present and analyse a decadal prediction system within the EC-Earth3 model contributing to the CMIP6 DCPP-A. The paper is organized as follows: section 2 provides a description of the EC-Earth3 forecast system, the initialisation approach considered, the skill evaluation metrics and the observational datasets used. In section 3, we characterize the predictive capacity for the surface temperatures and investigate the importance of the initialisation on surface temperatures,

upper ocean heat content and several interannual-to-decadal indices of climate variability, followed by an analysis of the predictive skill in the North Atlantic. This section illustrates that the low skill in the Central North Atlantic appears to be related to a strong initial shock. The final section summarizes the key results and conclusions of this study.

## 2 Data and Methodology

### 2.1 EC-Earth3 Decadal Forecast System

All experiments analysed in this study were performed with the CMIP6 version of the EC-Earth version 3 Atmosphere-Ocean General Circulation Model (AOGCM) in its standard resolution (Döscher and the EC-Earth Consortium, in prep.). Its atmospheric component is the Integrated Forecast System (IFS) from the European Centre for Medium-Range Weather Forecasts (ECMWF), cycle cy36r4, with a T255 horizontal resolution (grid size approximately 80km) and 91 vertical levels. The ocean component is the version 3.6 of the Nucleus for European Modelling of the Ocean (NEMO; Madec and the NEMO Team,

2016), which is itself composed of the ocean model OPA (Ocean PArallelise) and the Louvain-La-Neuve sea ice model (LIM3; Rousset et al., 2015), both run with an ORCA1 horizontal resolution (ca. 1° nominal resolution) and 75 vertical levels. The atmospheric and oceanic components are coupled through OASIS (Craig et al., 2017). The vegetation fields are prescribed and have been derived from an EC-Earth historical simulation coupled with the LPJ-GUESS dynamic vegetation model (LPJGuess; Smith et al., 2014).

Our decadal prediction system follows the CMIP6 DCPP-A protocol (Boer et al., 2016) and therefore consists of 10 member ensembles of 10-year long predictions initialised every year in November from 1960 to 2018 (referred to as PRED hereafter). To determine the impact of initialisation, PRED is compared with an ensemble of 15 CMIP6 historical simulations (1960-2015) (Eyring et al., 2016) continued with the SSP2-4.5 scenario simulations (2015-2100) (O'Neill et al., 2016) and performed with the same model version as PRED. These experiments (referred to as HIST hereafter) correspond to the CMIP6 members





(2,7,10,12,14,16-25), all the ones which were performed at the Barcelona Supercomputing Center (BSC). PRED and HIST are both performed with prescribed CMIP6 radiative forcing estimates (i.e., solar irradiance, and green house gas, anthropogenic aerosol and volcanic aerosol concentrations) for the historical period (1960-2014) and the SSP2-4.5 scenario on the subsequent years (Eyring et al., 2016).

In PRED, the different components (atmosphere, ocean and sea ice) have been initialised using full-field initialisation. The
atmospheric initial conditions have been interpolated from ERA-40 reanalysis outputs (Uppala et al., 2005) for the period 1960-1978 and from ERA-Interim (Dee et al., 2011) for the period 1979-2018. The ERA-Interim surface fields were replaced by the ERA-Interim/Land offline land surface reanalysis (Balsamo et al., 2015) driven by ERA-Interim surface fields and bias-corrected using precipitation from the Global Precipitation Climate Project (GPCP, Adler et al., 2018). The ocean and sea ice initial conditions come from a NEMO-LIM reconstruction, forced at the surface with fluxes from the Drakkar Forcing Set v5.2
(DFS5.2 Brodeau et al., 2010) up to 2015 and with ERA-Interim (Dee et al., 2011) thereafter. In this reconstruction, ocean temperature and salinity fields from the ORAS4 reanalysis (Mogensen et al., 2012) are assimilated through a standard surface nudging approach (e.g., Servonnat et al., 2015), using temperature and salinity restoring coefficients of $-40W/m^2/K$ and $-150mm/day$, respectively. Even if no direct assimilation of sea ice products is performed, the atmospheric surface fluxes combined with the surface ocean temperature nudging, are sufficient to bring the initial sea ice state close to observations (e.g.,
Guemas et al., 2014). Below the mixed layer, a Newtonian relaxation term is also applied to assimilate 3D ORAS4 temperature and salinity fields. For this, a relaxation timescale that increases monotonically with depth is used, which takes approximate values of 10 days at 1000m, 100 days at 3000m and 330 days at 5000m. Subsurface relaxation is applied everywhere except within the 3ºS–3ºN band to prevent spurious vertical velocity effects (Sanchez-Gomez et al., 2016).

To generate the 10 members of PRED different strategies are followed depending on the model component. The ensemble
spread for the atmospheric initial conditions is generated by adding infinitesimal random perturbations to the 3-dimensional temperature field. For the ocean and sea ice initial conditions, five different reconstructions are performed following the nudging strategy previously described, each assimilating one of the 5 members of ORAS4. The 5 ocean and sea-ice states generated are combined with two different atmospheric initial conditions each to produce the 10 ensemble members.

All the simulations completed in this study were performed in the supercomputer Marenostrum IV, hosted at the BSC, using
the Autosubmit workflow manager (Manubens-Gil et al., 2016), a Python toolbox specifically developed at the BSC to facilitate the production of experimental protocols with EC-Earth. This toolbox can easily handle experiments with different members, different start dates and different initial conditions. Autosubmit is hosted in the Gitlab repository of the BSC Earth Sciences Department (https://earth.bsc.es/gitlab/es/autosubmit). The scripts to run the model and all subsidiary tools are also included in the Gitlab repository under version control, and the tool that generates the perturbations saves the seed employed for each
member, both contributing to guarantee the reproducibility of the experiments within the maximum fidelity permitted by the model.

The raw model outputs were formatted following the Climate Model Output Rewriter (CMOR)/CMIP6 conventions to ensure efficient use and dissemination with the scientific community. This was done with 'ece2cmor' (https://github.com/EC-Earth/ece2cmor3), a Python tool for post-processing and cmorisation developed for EC-Earth3 which was implemented in the





Autosubmit workflow. After re-formatting, the model data was systematically quality checked with various tools to ensure no missing files and scientific validity. Both PRED and HIST experiments are published on the BSC data node of the Earth System Grid Federation (ESGF) where they are publicly available.

## 2.2 Observational Data for Verification

Various datasets are used as reference for estimating the forecast quality of the two EC-Earth3 ensembles HIST and PRED. To
evaluate surface temperature we use the gridded temperature anomaly products NASA GISTEMPv4 (Lenssen et al., 2019) and the Met Office HadCRUT4 (Morice et al., submitted). Both datasets combine near-surface air temperature (SAT) over land and sea surface temperature over the ocean (SST). For indices related to SST only, we use the Met Office HadISSTv3 (Kennedy et al., 2011). For upper-ocean heat content we use the Met Office EN4.2.1 gridded ocean temperature (Good et al., 2013). For comparing spatial fields with observations, EC-Earth3 predictions and historical simulations are re-gridded to the observational
grid in the case of the surface temperature variables corresponding to a 2ºx2º regular grid for NASA GISTEMP4 and a 5ºx5º regular grid for HadCRUT4. Ocean heat content is re-gridded to a 2ºx2º regular grid. Model simulations are masked in regions where and when observations have missing values. The regions with missing values in observations remain similar over the investigated period, especially for NASA GISTEMPv4.

## 2.3 Forecast Drift Adjustment and Verification Metrics

In the full-field initialisation approach, models are initialised close to the observed state and as the forecasts progress they experience a drift towards the (imperfect) model attractor. This drift needs to be corrected to prevent systematic errors in the prediction. To avoid drift-related effects, the evaluation of climate predictions against observations is performed in the anomaly space (e.g., Meehl et al., 2014). In this paper we use the 'mean drift correction' method which consists in computing the anomalies relative to the forecast time-dependent climatology, assuming that drift affects equally predictions from all start
dates (e.g., García-Serrano and Doblas-Reyes, 2012; Goddard et al., 2013).

Observed and HIST anomalies are computed with respect to their climatologies, in the case of HIST it is computed from the ensemble mean. All climatologies are computed for the common reference period 1970-2018. This is the longest period for which predictions at all forecast years (1 to 10) can be produced, and thus allows us to compute a climatology that is consistent across the whole forecast range. For forecast quality assessment purposes, we use the longest period available for
each forecast year (e.g. 1961-2018 for forecast year one, and 1970-2018 for forecast year ten), to produce in each case the most accurate estimate of the predictive skill. This implies that the skill of PRED and HIST may change with the forecast time as the verification period changes.

To measure the forecast quality we use the Anomaly Correlation Coefficient (ACC) and the Mean Square Skill Score (MSSS). The ACC measures the linear association between the predicted mean and the observations, but is insensitive to
the scaling. To determine the impact of the initialisation we compute the ACC differences between the decadal hindcasts and historical simulations. To determine whether these differences are statistically significant we follow the methodology developed by Siegert et al. (2017), which corrects for the independence assumption when two forecasts are strongly correlated. In





addition to the differences in ACC, to investigate the skill associated with the initialisation, we use the Mean Square Skill Score ($MSSS = 1 - (MSE\_PRED/MSE\_HIST)$; where MSE_PRED and MSE_HIST are the mean square errors between ob-

servation and PRED and HIST, respectively), which is described in detail in Goddard et al. (2013). The MSSS is a skill metric that allows us to compare the performance of PRED with respect to HIST. In the absence of a mean bias the MSSS can be divided into two terms, the correlation and the conditional bias, which provide different information. While the correlation takes into account the interannual variability and sign of linear trend, it is scaling invariant (i.e. independent of the signal amplitude). The conditional bias does consider the magnitude (amplitude) of the timeseries and the linear trend (Goddard et al., 2013). The

MSSS is thus a skill measure that depends on the correlation minus a penalisation for the (conditional) bias. The statistical significance of the MSSS is estimated using a random walk test following the methodology developed by DelSole and Tippett (2016).

To evaluate the predictive skill of PRED and HIST we compare them to a persistence forecast. The persistence forecast was computed by taking the monthly or annual anomaly at the time of initialisation and persisting it for all forecast times (e.g.

Yeager et al., 2018). Therefore the correlation skill for the persistence forecast is equivalent to a lag auto-regressive model.

The spread-error-ratio (SER; Ho et al., 2013) has been used as an indicator of the forecast reliability, which is defined as the ratio between the mean intra-ensemble standard deviation and the root-mean-square error of the forecast ensemble mean. When the SER is greater (lower) than one, the ensemble is over-dispersed (under-dispersed) and the predictions will be under-confident (over-confident).

For data retrieval, loading, processing and calculation of verification measures 'startR' and 's2dverification' (Manubens et al., 2018) R libraries have been used, both developed at the BSC.

## 2.4 Climate Indices and Diagnostics

Several climate indices are used to assess the ability to predict both global and regional multi-annual variability.

Global mean surface temperature (GMST) is derived by blending SST over ocean and SAT temperatures over land. This

allows for a consistent comparison with the aforementioned observational datasets NASA GISTEMPv4 and HadCRUT4. The use of GMST over global surface air temperature at 2m height (GSAT) is particularly favourable when assessing long-term climate trends (e.g., Richardson et al., 2018).

In the Pacific ocean, to distinguish between seasonal-to-interannual and decadal variability, we look at the ENSO and IPO respectively. For ENSO we use the NINO3.4 index, which is defined as the area weighted average over the region: 5°N-5°S and

170°W-120°W. For the IPO we use the Tripolar Pacific Index (Henley et al., 2015), which corresponds to the average of SST anomalies over the central equatorial Pacific (region 2: 10°S–10°N, 170°E–90°W) minus the average of the SST anomalies in the Northwest (region 1: 25°N–45°N, 140°E–145°W) and Southwest Pacific (region 3: 50°S–15°S, 150°E–160°W). To describe the decadal variability over the Atlantic basin, we use the AMV definition from Trenberth and Shea (2006). The AMV is calculated as the spatial average of SST anomalies over the North Atlantic (Equator-60°N and 80°–0°W) minus the spatial

average of SST anomalies averaged from 60°S to 60°N (Trenberth and Shea, 2006; Doblas-Reyes et al., 2013). In addition to the AMV, we also compute the Subpolar North Atlantic (50–65ºN,60–10ºW) ocean heat content in the upper 300m (referred





to as SPNA-OHC300 hereafter). Since the IPO, AMV and SPNA-OHC300 are decadal modes of variability, we filter out part of the interannual variability by considering four-year temporal averages along the forecast time (i.e. forecast years 1-4, 2-5, 3-6...) for these indices.

Likewise, density has been computed using the International Equation of State of seawater (EOS-80) referred to the level of 2000 dbar (sigma-2), a level that represents better the deep water properties. In addition, the contributions of temperature (sigma-T) and salinity (sigma-S) to density were computed using the thermal expansion and haline contraction coefficients, which were themselves estimated as the density change (in the EOS-80 equation) associated with a small increase in temperature (0.02 °C) and salinity (0.01 psu), respectively.

All ocean diagnostics have been computed using 'Earthdiagnostics', a python-based package developed at the BSC.

## 3    Results

### 3.1    Characterising the Predictive Capacity of Surface Temperature

#### 3.1.1    Global Mean Surface Temperature

First we compare the predicted GMST evolution for different forecast periods (Figure 1), estimated by combining SAT over
land and SST over the ocean (cf. Data and Methodology). PRED reproduces the observed variability and shows very high ACC skill values: 0.96, 0.97 and 0.95 for forecast year one, years 1-5 and years 6-10 respectively (similar values are obtained when comparing to other observational products like HadCRUT4). As expected, HIST does not capture most of the interannual variability (Figure 1), since it is largely of internal origin and therefore averages out by construction. Nevertheless, HIST shows very high skill (0.94, 0.96 and 0.95 for forecast year one, years 1-5 and years 6-10 respectively) associated with the
global warming trend and the cooling episodes in response to the volcanic eruptions of Agung (1963), El Chichon (1982) and Pinatubo (1991). The differences in ACC skill between PRED and HIST are not statistically significant, indicating that the high skill of PRED is mainly associated with the external forcings. When the simulations are detrended (a simple attempt to remove the warming trend) PRED shows higher ACC skill values than HIST, revealing the benefit of the initialisation, especially for the earlier forecast years (ACC values 0.75 and 0.5 in forecast year one for PRED and HIST, respectively).

Comparing the intra-ensemble spread of PRED and HIST (shown by the Box-Whisker for PRED and shading for HIST in Figure 1) shows that PRED has considerably smaller spread even on the longer forecast times. For example the mean intra-ensemble standard deviation of PRED is 0.05K while for HIST is 0.20K for the first forecast year. This is probably due to the initialisation of PRED, which phases simulated internal and observed variability and may also help to correct systematic errors in model response to external forcing (e.g., Doblas-Reyes et al., 2013). This difference in spread remains equivalent when the
ensemble size of HIST is reduced to 10 members, i.e. the same ensemble size of PRED.



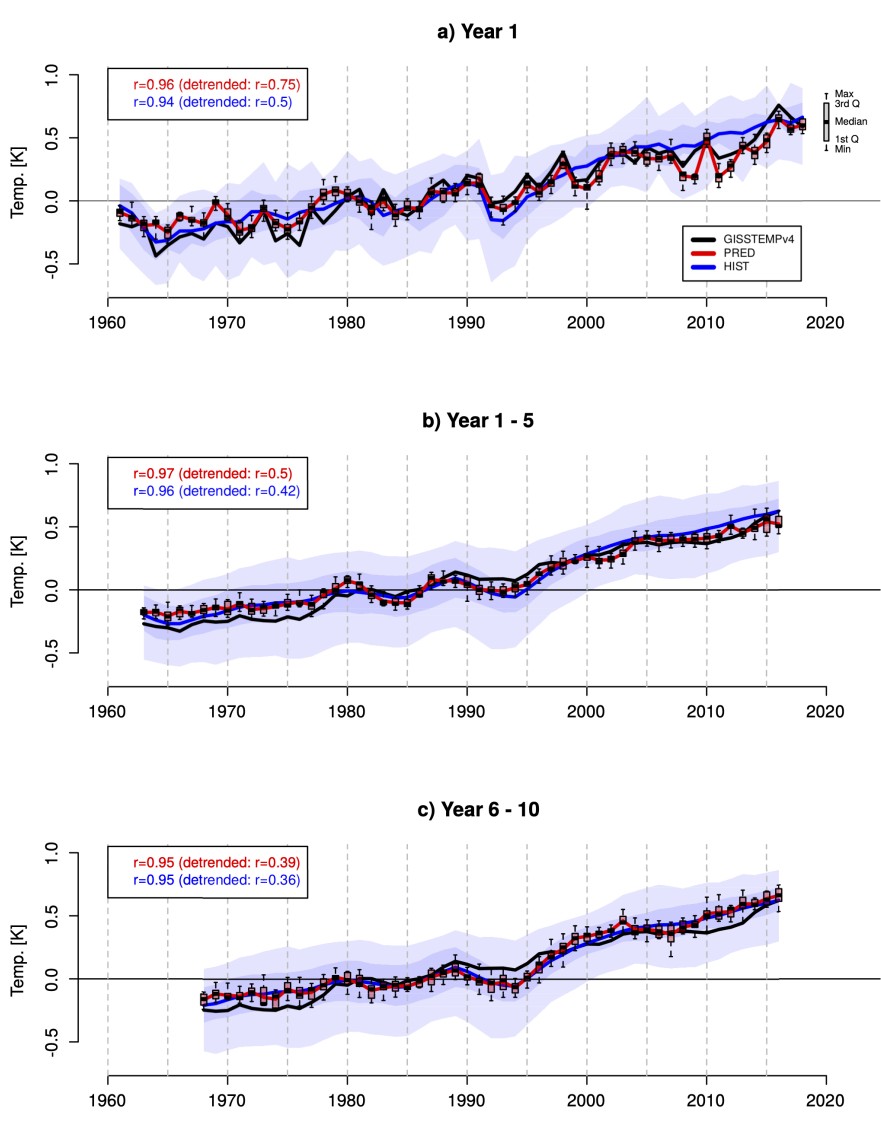

**Figure 1.** GMST annual mean anomaly timeseries (K) for PRED, HIST (historical+SSP2-4.5) and GISTEMPv4 for a) forecast years 1, b) forecast years 1-5 and c) forecast years 6-10. The anomalies cover the period 1961-2018 and are referenced to the 1971-2000 mean. Multi-year means (panels b and c) are plotted on the central year (e.g., 2000 represents the values from 1998-2002 in b and c). For a fair comparison with observations, PRED and HIST have been masked where and when GISTEMPv4 has missing values. The intra-ensemble spread for PRED and HIST is represented by the box-and-whisker plots and shading respectively. The ACC for PRED and HIST is shown in the top left part of each panel, including in brackets the ACC after removing a linear trend from the timeseries. All ACC values are statistically significant at the 95% level.





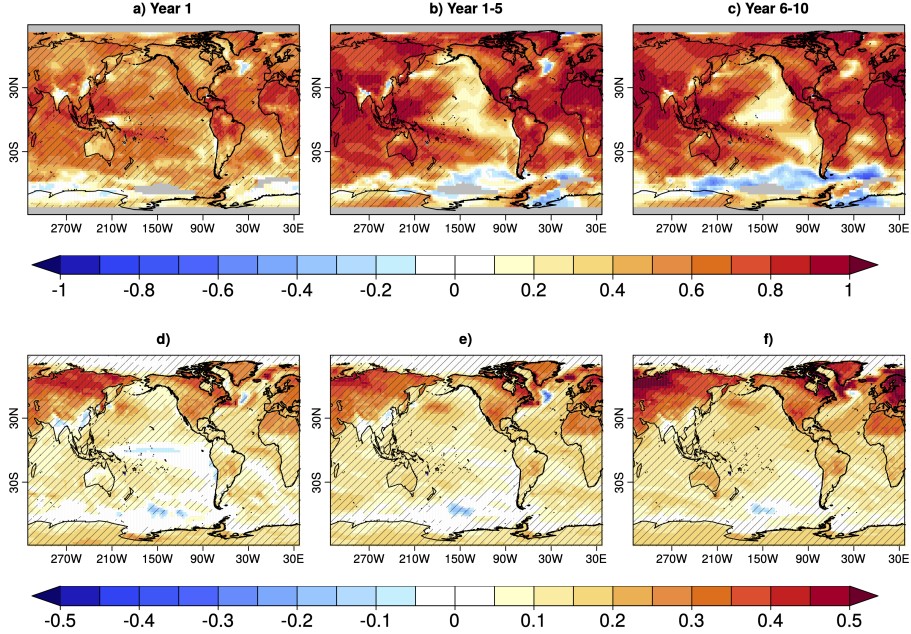

**Figure 2.** Top row: Anomaly correlation coefficient (ACC) for surface temperature in PRED for a) forecast years one, b) forecast years 1-5 and c) forecast years 6-10. The ACC is computed between the model ensemble mean of the blended SAT (over land) and SST (over the ocean) fields and GISTEMPv4. Hatching indicates areas that are significant at the 95% confidence level. Missing values in observations are masked in grey. Bottom row: surface temperature linear trends [in K/decade] for PRED for d) forecast years one, e) forecast years 1-5 and f) forecast years 6-10. Hatching indicates a trend that is statistically different from zero at the 95% confidence level. Both ACC and trends are computed for annual mean values of all available years for the respective forecast period referenced to the 1970-2018 climatology.

### 3.1.2 Added Value of Initialisation at the Regional Scale

At the regional level, PRED shows high skill at predicting surface temperature at different forecast ranges (Figure 2a-c), as expected from the presence of long-term warming trends (Figure 2d-f). In the first forecast year, most regions show significant skill, with a few exceptions such as the central Subpolar North Atlantic, some regions of Asia, Australia and the Southern Ocean, where the simulated trends are small and mostly not statistically significant (Figure 2d). In contrast, the Eastern Pacific shows no significant trends but does show significant skill in the first forecast year associated with the initialisation of ENSO. On longer time-scales (forecast ranges 1-5 and 6-10 years) PRED also shows significant skill in many regions worldwide with greater ACC values compared to forecast year one (Figure 2b and c). This is probably a consequence of considering 5-year averages for validation, which reduces the influence of interannual variability. There is however an important degradation of the skill in some regions for these forecast ranges, in particular in the Eastern Tropical Pacific where the model might not be representing the correct ENSO low frequency variability, and in the North Pacific where generally low levels of skill have been related to model biases in ocean mixing processes (Guemas et al., 2012). Comparing the forecast periods for years 1-5 and





6-10, two major differences are apparent. First, in the Southern Ocean, skill degrades dramatically with forecast time, which is probably associated to the development of a warm bias due to the incorrect representation of cloud feedbacks in the region

(Hyder et al., 2018). Second, the central Subpolar North Atlantic seems to exhibit an opposite change in skill, from negative ACC values during the first 5 forecast years to positive but insignificant ACC values in the five following years, which might reflect the recovery from an initial shock that affects the North Atlantic. This shock might be responsible for the strong negative trends over the regions in forecast years 1-5, which are substantially reduced in forecast years 6-10 (Figure 2e,f). This will be further discussed in 3.3.

To determine whether there is a benefit of the initialisation, we compute the MSSS of PRED considering HIST as the baseline (Figure 3a-c). To aid in the interpretation of the MSSS, we also show the ACC difference between PRED and HIST (Figure 3d-f) and the conditional bias difference (Figure 3g-i). In the first forecast year, the MSSS shows added value from initialisation mostly in the Pacific Ocean, the neighboring region of the Southern Ocean, the eastern Subpolar North Atlantic and the Atlantic Subtropical areas (Figure 3a). An inspection of the ACC and conditional bias difference maps (PRED-HIST)

reveals that for the first forecast year, the positive MSSS values (which are indicative of added value of initialisation) are mostly associated with increased ACC values in PRED (i.e. the hindcasts reproduce better the observed variability), while the Subtropical Atlantic is the only region where a positive MSSS is associated with a reduction in the conditional bias (i.e. a better representation of the linear trend).

MSSS values become predominantly negative for forecast years 1 to 5 (Figure 3b,e,h) and only a few regions benefit from

the initialisation, such as the Tropical Pacific (Figure 3b) due to a reduction of the conditional bias (Figure 3h). The rest of the Pacific and most of the Atlantic basin show negative MSSS values (Figure 3b). In these regions, PRED has generally lower ACC values than HIST, and a larger conditional bias (Figure 3e,h). At longer forecast times (6-10 years), positive MSSS values are almost exclusive to the Eastern tropical South Pacific and tropical South Atlantic, where the conditional bias of the forecast is reduced (Figure 3i), indicating a better representation of the long term trend in PRED with respect to HIST. By

contrast, negative MSSS values span across most of the North Atlantic, and reach the Eurasian continent (Figure 3c), mostly associated with a larger conditional bias in PRED compared to HIST 3i). The ACC differences are also predominantly negative in the Subpolar North Atlantic, a region in which many prediction systems exhibit prolonged added value of initialisation (e.g., Robson et al., 2018; Yeager et al., 2018). This suggests that some regional key physical processes (e.g., the gyre and overturning ocean circulations) might not be well represented in EC-Earth3 predictions.

To complement the analysis of surface temperature, we also consider the upper ocean heat content, a quantity that better represents the thermal inertia of the ocean and a source of decadal variability and predictability for the surface climate (e.g., Meehl et al., 2014; Yeager et al., 2018). As previously shown for surface temperature, in the first forecast year high and significant ACC values are obtained in all major basins for the ocean heat content in the upper 300m (referred to as OHC300 hereafter; Figure 4a). A region of negative skill values is evident over the central Subpolar North Atlantic as for the surface

temperatures (Figure 2a). Forecast years 1-5 and 6-10 show that the skill in the Tropical and Eastern Pacific is lost as is for the some regions in the Atlantic and Pacific sectors of the Southern ocean (Figure 4b and c). As for surface temperature, the skill in the central Subpolar North Atlantic moderately improves in forecast years 6-10 with respect to forecast years 1-5.



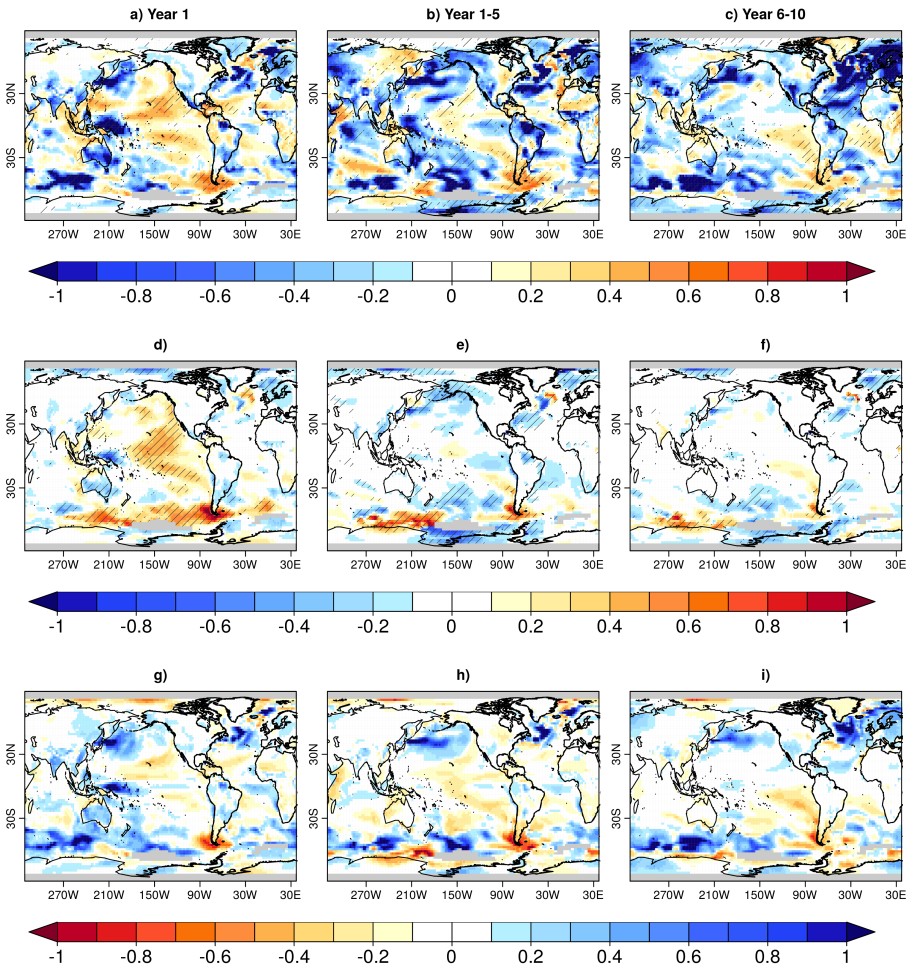

**Figure 3.** Annual mean surface temperature (SAT and SST blend) skill comparison and bias differences between PRED and HIST, computed with the GISTEMPv4 observations. The top row shows the MSSS of PRED with the HIST as the reference forecast (see section 2.3) for a) forecast year one, b) forecast years 1-5 and c) forecast years 6-10. Hatching indicates where the MSSS is significant using a random walk test (see section 2.3). The middle row shows the ACC difference between PRED and HIST for d) forecast year one, e) forecast years 1-5 and f) forecast years 6-10. Hatching indicates a significant difference in ACC between PRED and HIST based on the methodology developed by Siegert et al. (2017) (see section 2.3) at the 95% confidence level. The bottom row shows the difference in absolute values of the conditional bias between PRED and HIST for g) forecast year one, h) forecast years 1-5 and i) forecast years 6-10. Annual mean anomalies are computed masking PRED and HIST with the GISTEMPv4 missing values (masked in grey) and using the common climatology period 1970-2018.



Comparing the ACC difference between the PRED and HIST ensembles reveals that the initialisation increases the forecast skill of OHC300 in the eastern Subpolar North Atlantic in all the three forecast times (and ranges) considered (Figure 4d-f). A result that is consistent with other forecast systems (e.g., Robson et al., 2018; Yeager et al., 2018). The Pacific Ocean shows significantly improved skill from initialisation basin wide in the first forecast year (i.e. ENSO), but for forecast years 1-5 and 6-10 the added value of initialisation is limited to parts of the Eastern subtropical Pacific. The initialisation improves the skill in most of the Indian Ocean at all forecast times considered (although it is not statistically significant for the forecast range 6-10), consistent with previous studies showing the high skill of decadal predictions in this region (Guemas et al., 2013).

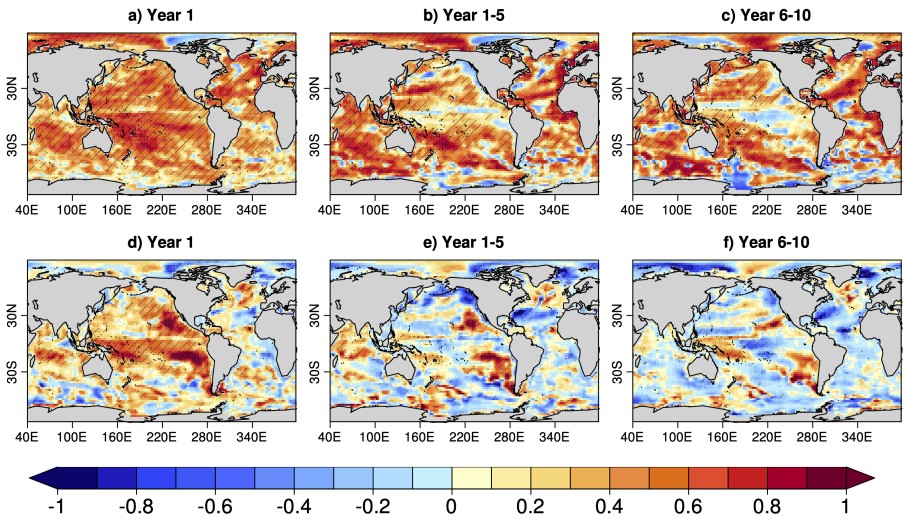

**Figure 4.** Upper 300m OHC ACC of PRED computed with the EN4 observations for a) forecast year one, b) forecast years 1-5 and c) forecast years 6-10. The impact of initialisation is shown as the difference in ACC between PRED and HIST for d) forecast year one, e) forecast years 1-5 and f) forecast years 6-10. In panels a-c, the hatching indicates the statistical significance of the correlation at the 95% confidence level. For panels d-f hatching indicates the regions where the difference in correlation between HIST and PRED are statistically significant at 95% confidence level.

## 3.2 Skill for the Main Ocean Modes of Variability

We further evaluate the predictive capabilities of the EC-Earth3 PRED experiment by considering the skill for predicting several modes of interannual to decadal variability (Figure 5). In the Pacific Ocean, ENSO is the main mode of climate variability on seasonal to interannual time-scales, and can help the predictive capacity worldwide through its well-known climate impacts (e.g., McPhaden et al., 2006; Yuan et al., 2018). Figure 5a shows that PRED captures the year-to-year evolution of the observed ENSO. This is confirmed by the high ACC values during the first four forecast months (ACC>0.9), which are followed by a typical loss of skill that many dynamical forecast systems experience during the spring season (i.e. the spring barrier; Webster and Yang, 1992) and by a recovery in summer through the next winter, in which ACC values remain positive and significant



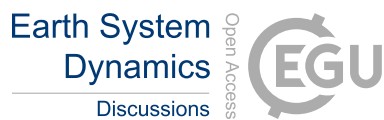

(Figure 5b). Added value for ENSO due to initialisation is evident up to the second forecast year, as indicated by the positive MSSS values and the statistically significant difference in ACC values between PRED and HIST (Figure 5b). The spread-error-
ratio, however, reveals that the ENSO predictions are overconfident for the first few months, in contrast to HIST which tends to be underconfident. On decadal time-scales, the dominant mode of climate variability in the Pacific basin is the IPO. Figure 5e shows that neither PRED nor HIST are capable of skillfully predicting it, a lack of skill that has been documented in many other prediction systems (e.g., Doblas-Reyes et al., 2013). Nonetheless, initialisation does seem to improve the reliability of the IPO 5f.

In the Atlantic Ocean, the AMV is the dominant mode of decadal climate variability, and has been linked to several climate impacts over Europe, North America and the Sahel (Zhang and Delwoth, 2006; Sutton and Dong, 2012; Ruprich-Robert et al., 2017, 2018) and to Atlantic tropical cyclones (e.g. Caron et al., 2015, 2018). Both PRED and HIST are capable of skillfully predicting the AMV, and do better than a persistence forecast (except for the forecast range 1-4 years in HIST), as shown by the ACC (Figure 5h). PRED however, is consistently better than HIST as shown by the MSSS, even though the ACC differences
are not statistically significant. The spread-error-ratio shows that initialisation improves the reliability of the AMV predictions at all forecast ranges (Figure 5i), since the historical simulations are overdispersive probably due to excessive intra-ensemble spread as previously described in section 3.1.1.

Since the Subpolar North Atlantic has been shown to be a region where forecast systems exhibit skill on decadal times scales, we analyse the SPNA-OHC300 index (see Section 2.4). PRED exhibits a lack of skill in this index up to forecast range 4-7,
with significant ACC values emerging for longer forecast ranges, coinciding with the time in which the system outperforms persistence. ACC values in the HIST ensemble (which are not statistically different from those in PRED) also increase with forecast time. In HIST, this is due to the fact that the skill for each forecast range is computed over a different verification period, the same one used for PRED, which used for each forecast range the longest period available. For the longest forecast ranges [e.g., 7-10], the first start dates cannot be used, excluding some of the earliest years [e.g., 1960-1966], for which the warming
trend was less prominent, thus producing an artificial increase in skill for longer forecast times. Repeating the calculations for PRED over a common verification period to all forecast ranges (i.e. 1970-2018) reveals that lower skill values are still present for the first forecast years (see Supplementary Figure 1), which suggests that the differences in skill with forecast time are not due to differences in the verification period but to other causes, for example some potential initialisation shocks from which the system is recovering some years later. This possibility is discussed in the next sub-section. As for the AMV, the initialisation
improves the reliability of the PRED and shows the HIST intra-ensemble spread may be too large and therefore under-confident (Figure 5l).

### 3.3 Understanding the Limited Predictive Skill in the Subpolar North Atlantic

In the previous section we have shown an overall detrimental effect of initialisation in the EC-Earth3 predictions over some regions of the North Atlantic at all forecast ranges(Figure 3), leading to lack of predictive skill in the specific case of the central
Subpolar North Atlantic as shown by the ACC maps of surface temperature (Figure 3) and upper ocean heat content (Figures 4 and 5). Decadal variability in the Subpolar North Atlantic is highly influenced by the changes in the ocean circulation, both





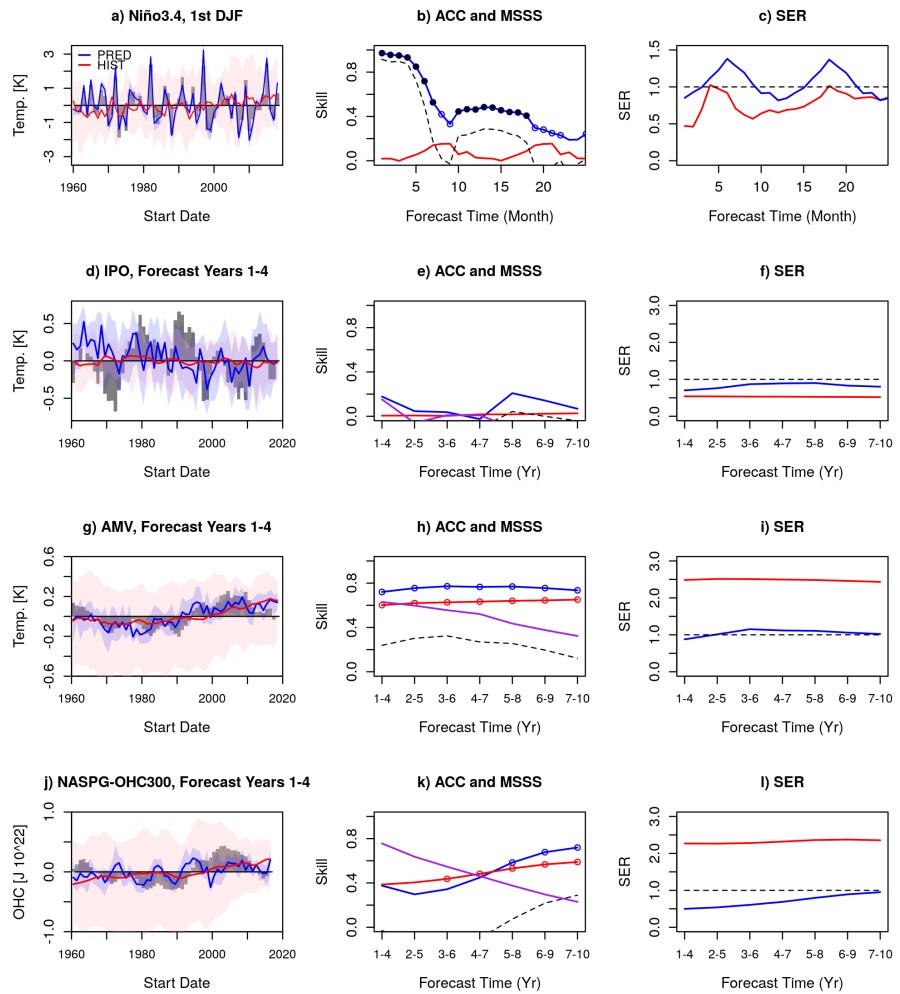

**Figure 5.** Skill of the selected modes of ocean variability: a-c) ENSO, d-f) IPO, g-i) AMV and j-l) SPNA-OHC300. The first column shows the observed (grey bars) and predicted (PRED in blue, HIST in red) time series of the indices. The ensemble means are represented with lines and the ensemble spread with coloured envelops. The first column shows the ENSO index for the first winter (DJF), while for the other indices the average of the first 4 forecast years is shown. The second column shows the ACC of PRED (blue) and HIST (red), the MSSS of PRED considering HIST the baseline prediction (black dashed line) and a persistence forecast (purple). Statistically significant ACC values (at the 95% confidence level) are shown as empty circles. ACC differences that are statistically significant (at the 95% confidence level) between the PRED and HIST are shown as filled circles. The third column shows the spread-error-ratio of PRED (blue) and HIST (red).





meridional and barotropic (e.g., Ortega et al., 2017). To understand the role of the ocean circulation we analyse the evolution of the Atlantic Meridional Overturning Circulation at 45ºN (defined as the overturning stream-function value at 45ºN and at 1000m depth; referred to as AMOC45N hereafter) and North Atlantic Subpolar Gyre Strength Index (NASPG, defined as

the regional average of the barotropic stream-function in the Labrador Sea [52-65ºN, 58-43ºW], multiplied by minus one to make the values positive to aid the comparison) in PRED and HIST (Figure 6a and b). Additionally we include the ocean-only reconstruction from which the initial conditions are obtained (referred to as RECON hereafter) to determine how the predictions depart from the initial conditions. Actual model values are used to illustrate how the forecast drift develops. The mean forecast drift is also shown for completeness, estimated as the climatological value as a function of forecast time (Figure

6c and d). Figure 6 shows that decadal changes in the AMOC and NASPG are highly correlated (e.g. R=0.8 in RECON), a feature that has been shown in previous studies (Ortega et al., 2017).

Comparing PRED and RECON allows us to identify several interesting features. In the first forecast year the predicted AMOC45N is of equal value with respect to RECON, while for the NASPG index the predicted values tend to be weaker (Figure 6). As the forecast evolve and the model transitions towards its free-running attractor both indices diverge from RECON and

experience a pronounced weakening. By forecast year ten, the indices in PRED reach a weaker mean state than in HIST (black dashed lines in Figure 6c and d, respectively). These differences between PRED and HIST suggest either that the forecasts in PRED need to be run for longer to reach its attractor (HIST) (e.g., Sanchez-Gomez et al., 2016), or that the attractor of PRED is different from HIST.

For both indices, we also note a clear difference in the way the forecast transitions to the model attractor before and after

year 2000. In the first 30 start dates, the AMOC45N and NASPG in PRED start at stronger values than HIST (c.f. RECON values in Figure 6a and b), and the individual predictions exhibit a fast decline that surpasses the HIST mean state. In year 1995 of RECON, both indices experience a sharp decrease and eventually stabilise around a substantially lower mean state, a transition that has been shown to be partly predictable in previous studies (Robson et al., 2012; Yeager et al., 2012; Msadek et al., 2014). Due to this lower initial state, all predictions after the year 2000 start much closer to the HIST mean state and the

PRED attractor. In consequence, the drift in PRED is smoother. The fact that there are two distinct periods in which the model drifts in different ways (Figures 6c and d) may compromise the applicability of the drift correction methods used to compute the forecast anomalies, which assumes a stationary forecast drift. This is particularly evident for the AMOC45N, which shows important differences in the PRED climatologies during the first three forecast years when the climatologies are computed for the time periods preceding and following the year 2000 (Figure 6c red and blue lines, respectively). These period-sensitive

climatological differences are less pronounced for the NASPG (Figure 6d).

To understand why the AMOC45 and NASPG are not stabilising in the predictions around the mean HIST state, we focus on the Labrador Sea. The Labrador Sea is a key region of deep water formation, in which climate models show limitations to represent realistic oceanic convection, which can happen too often, too deep, or can be completely absent in some cases (Heuzé, 2017). Figure 7a shows the mixed layer depth (MLD) evolution in the Labrador Sea, a proxy for the convection activity in this

region. The MLD index is computed as the average of February-March-April, the months with the deepest mixing. In PRED, MLD systematically collapses within the first three forecast years, which is in stark contrast with the typical behaviour in



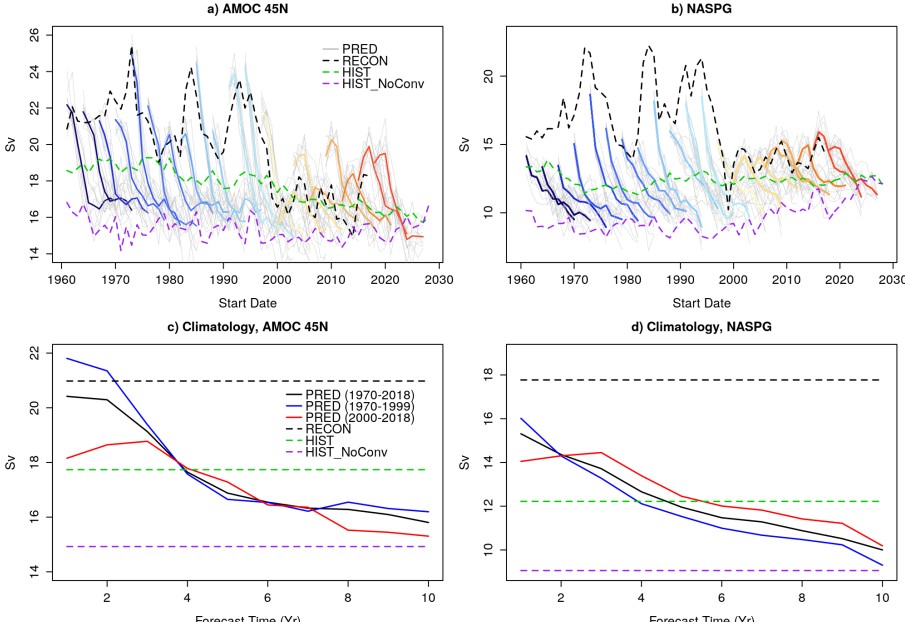

**Figure 6.** Evolution of the a) AMOC45N and b) NASPG in the raw forecasts, historical ensemble and reconstruction. Ensemble mean forecasts (10 members) of PRED are shown from blue to red every 3 startdates, with individual ensemble members shown in grey. The ensemble mean RECON (5 members) is shown with the black dashed line. The ensemble mean of all HIST (15 members) is shown in green, and the ensemble mean of the HIST members that do not exhibit convection are shown in purple. Panels c) and d) show the PRED climatological values as a function of forecast time for the AMOC45N and NASPG, respectively. Three time periods are considered for PRED: in black the climatology is for the period of 1970-2018, in red for the period 1970-2000 and in blue for 2000-2018. The black, green, and purple dashed lines indicate the climatology computed over the 1970-2018 period for RECON, all HIST members, and HIST members that do not show convection, respectively.

the HIST ensemble, in which deep convection happens regularly. In the HIST ensemble mean, Labrador convection remains active throughout the whole period although it exhibits a long-term weakening trend, consistent with the increase in local stratification caused by the externally forced ocean surface warming. The Labrador MLD index also allows us to identify three
HIST members with a distinct evolution from the rest, characterised by no convection during most of the historical period with slight increase from 2005 onward (purple line in Figure 7a). These simulations have a remarkable similarity with the state towards which PRED appears to be drifting. The ensemble mean of these three HIST members is also compatible with the AMOC45N and NASPG states at the end of the forecasts (purple lines in Figure 6), suggesting that the attractor reached by PRED (by forecast year 10) is associated with a suppressed Labrador Sea convection state. Note also that in the first forecast
year of PRED the Labrador Sea MLD is stronger than in RECON. All of the above suggests the existence of an initial shock in PRED, which initially boosts convection and subsequently brings the model towards a non-convective state.





Other key indices are also affected by the Labrador Sea convection collapse in PRED. For example, we see that sea ice grows to occupy the whole Labrador Sea as soon as convection ceases (Figure 7b and e). Like for the MLD, the sea ice extent of the HIST members with no convection is remarkably similar, while convection in the other members keeps a relatively reduced

sea ice coverage. The western SPNA-OHC300 (50–65ºN,60–30ºW) also seems to experience an initialisation shock as shown in Figure 7c; the PRED climatological value at forecast time 1 year is lower than in the RECON climatology (Figure 7f). In forecast years 2-3, this index tends to increase, approaching the HIST mean state, which is higher than in RECON. However, this trajectory changes drastically after forecast year 3 (7f), and a quick cooling begins towards the no-convection HIST state. This sudden change could be explained by a delayed response to the convection collapse in the predictions, which is expected

to drive a weakening of the SPG intensity by decreasing the density of its inner core and its associated geostrophic current (Levermann and Born, 2007). For all these indices we note again that their climatological drifts seem non-stationary (Figure 7d, e and f), and that predictions started after the year 2000 might not be well bias corrected. This, together with the strong initial shock, could explain the low (negative) predictive skill in the western SPNA region (Figures 2 and 4).

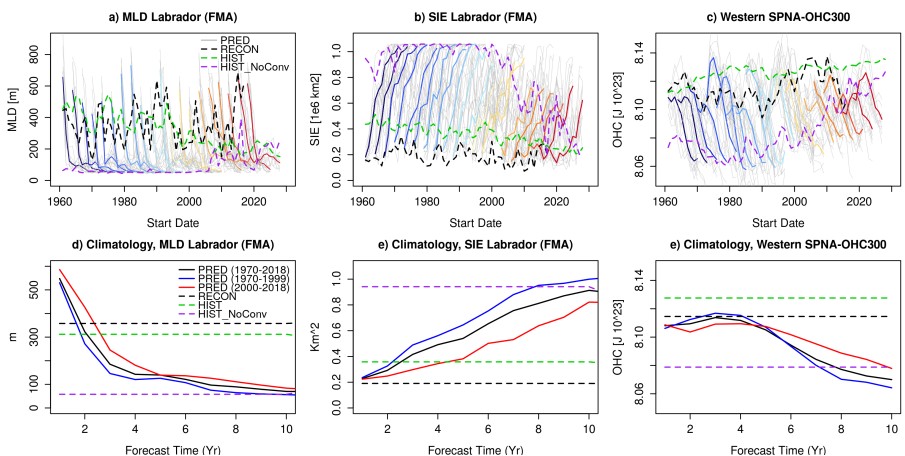

**Figure 7.** The same as in Fig. 6 but for the MLD in the Labrador Sea February-March-April, the SIE in the Labrador Sea during the same months and the western SPNA-OHC300 annual mean.

### 3.4    Insights on the Labrador Sea Initialisation Shock and Drift

Full-field initialisation can sometimes produce strong initialisation shocks and drifts, as the climate model adjusts from an initial state that might be substantially far from its attractor (e.g., Sanchez-Gomez et al., 2016). This section focuses on Labrador Sea Convection, for which Figure 7a shows a clear adjustment marked by an initial increase and a subsequent decline. Both aspects of the predicted Labrador Sea evolution are investigated separately. We focus on the preconditioning role of Labrador Sea density stratification on convection, and investigate the role of temperature and salinity, two variables that might be expe-

riencing a different initialisation adjustment and forecast drift over the region.





The initial enhancement of convection is explored in Figure 8, describing the evolution of stratification in the Labrador Sea the first five months of the forecast (Nov-Mar). At the time of initialisation (Nov), the density profiles of PRED and RECON are almost identical (dark red and blue lines in Figure 8a). Differences start to emerge in the subsequent forecast months, in which their density stratification weakens at a different pace, with PRED becoming more weakly stratified and therefore more

favorable to deep convection. HIST (green lines in 8a) also shows a similar tendency to reduce density stratification from November to March, although the Labrador Sea density remains more strongly stratified than in PRED and RECON, which would explain why convection is also weaker. By considering the temperature and salinity contributions to density stratification (sigma-T and sigma-S, figure 8b and c) we find that even though the overall density structure is dominated by salinity, with temperature largely opposing the mean density stratification, the major differences between PRED and RECON occur in the

sigma-T profile and are more notable at the surface. During the first 5 months of the forecast, sigma-T fully accounts for the differences between PRED and RECON in Labrador Sea density (e.g., 0.038 kg/m3 at the surface by March), with virtually no differences arising in sigma-S (0.005 kg/m3), which fails to counterbalance the temperature driven changes. As a result, the destabilising role of temperature on density stratification in the deep convection months is stronger in PRED than in HIST, promoting deeper convection.

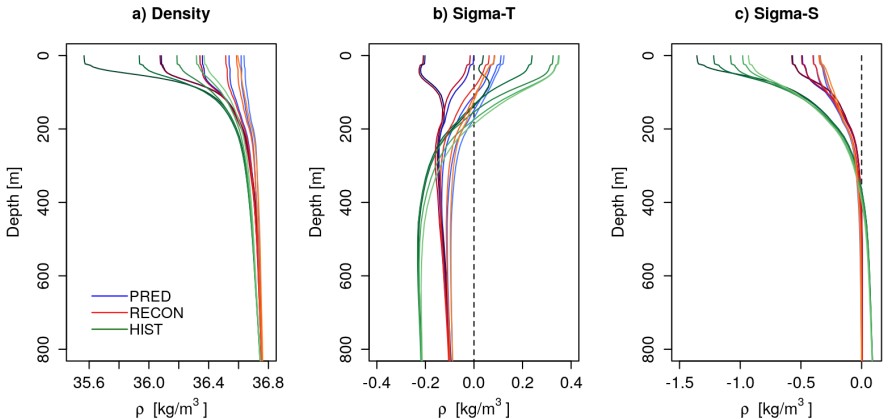

**Figure 8.** Labrador sea a) density, b) sigma-T and c) sigma-S climatological profiles in the first five forecast months (November to March) of PRED, and the equivalent calendar months of HIST and RECON. The colour intensity of the profiles from dark to light refers to increasing the forecast month.

To understand why Labrador Sea stratification diverges from RECON to PRED in the first forecast months we inspect the local surface restoring fluxes in the former, which, on average, are indicative of systematic model biases in the ocean component. In RECON, the heat flux restoring term is consistently positive and contributes thus to maintain a warmer surface in these months of deep convection (Figure 9). These fluxes are not present in PRED because the simulation are fully coupled, which will quickly adjust to a new free-running state with a colder upper Labrador Sea, explaining in this way the relative

surface cooling (and associated weakening of density stratification) with respect to RECON (Figure 8). Similarly, the freshwater





fluxes from the salinity restoring term are also positive in the Labrador Sea, and contribute to keep a fresher (and lighter) surface in RECON than in PRED. Its effect, however, appears to be small in magnitude as no remarkable differences emerge in sigma-S between RECON and PRED.

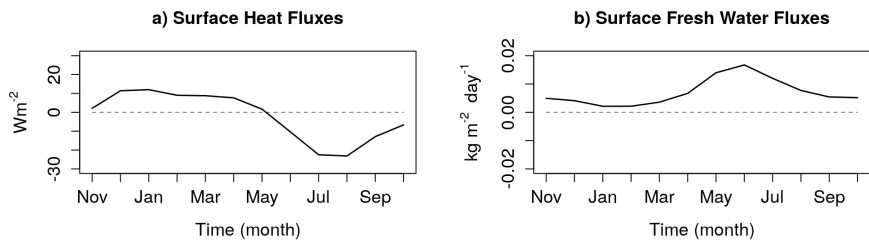

**Figure 9.** Labrador Sea [52-65ºN, 58-43ºW] monthly climatology of the nudging correction fluxes in RECON of a) heat and b) freshwater.

After better understanding the process behind the initial shock in the first winter, we now investigate the origin of the weak-
ening in the Labrador Sea convection after the first forecast year of PRED. Again we analyse the evolution of the Labrador Sea density profile in PRED, but for each convective season (FMA) as a function of the forecast year (Figure 10). The corresponding profiles for RECON and the ensemble members of HIST are included to contextualise the predictions. The sigma-T and sigma-S profiles are also shown to disentangle the contributions from temperature and salinity to density. After the first forecast FMA (darkest blue line in Figure 10), for which we showed a decrease in stratification that favoured deeper convection
with respect to RECON, the density stratification becomes increasingly stronger with forecast time. This evolution is explained by the changes in salinity (Figure 10c), as temperature contributes to decrease stratification at all forecast ranges (Figure 10b). In the second forecast FMA, density stratification in PRED is already stronger than in RECON (red line in Figure 10a). By the third forecast FMA it becomes stronger than in most of the HIST members with active convection (green lines), and by the sixth FMA it is already higher than in all of them. Interestingly, the stratification of sigma-S is not particularly different
in PRED than in the HIST ensemble members with convection, which suggests once again that the counterbalancing effect of sigma-T is important to understand the absence of convection in the forecasts. By the tenth (last) forecast FMA (lightest blue line in Figure 10) the density stratification is remarkably similar to that in the HIST ensemble members without convection. This may suggest that PRED is stabilising around this particular HIST state. However, this hypothesis is contradicted by the vertical profiles of sigma-T and sigma-S, that in the final forecast FMA appear to be more comparable to the HIST members
with convection. It is therefore possible that the forecast drift is bringing PRED to a different equilibrium state than in HIST.

To investigate the model drift in the Labrador Sea and how it affects its stratification figure 10 shows the scatter-plots of the climatological Labrador Sea FMA temperature and salinity both at the surface and at 500m of depth (Figure 11). At the surface, the mean temperature and salinity for the first forecast FMA remain close to those in RECON, as well as to the values in several HIST members with active convection, all placed along the same isopycnal ($27kg/m^3$). With increasing forecast
time, PRED drifts towards a state with lower temperature and fresher conditions, the same one of the HIST members with no convection. During this transition, the surface in PRED also becomes lighter, contributing to increase the stratification in the





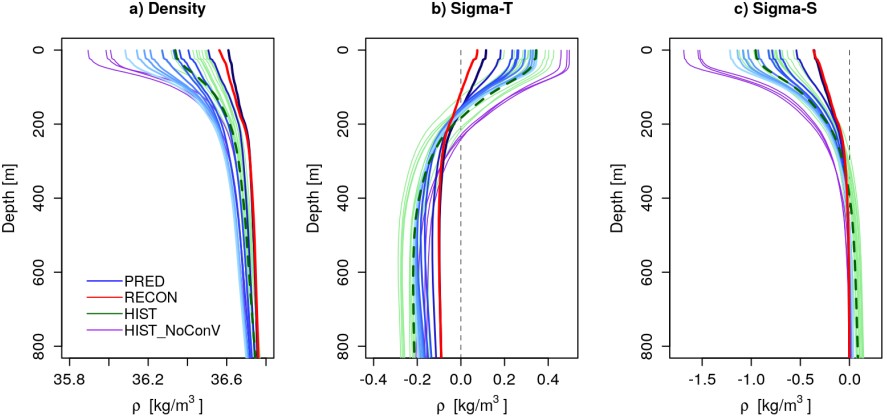

**Figure 10.** Labrador sea a) density, b) sigma-T and c) sigma-S climatological profiles for the convection season (February-March-April) in PRED, RECON and HIST. In PRED, the intensity of the blue lines is used to represent the changing forecast time, with the darkest blue line corresponding to the first forecast year, and the lighter blue line to the tenth. The HIST members have been divided into two sub-ensembles, those with and without convection in the Labrador Sea, HIST (green lines) and HIST-NoConv (purple lines). The green dashed line is the HIST ensemble mean using all members.

region. Important differences are observed in the subsurface (i.e. 500 m). For example, unlike for the surface, all the HIST members (i.e. the convective and non-convective ones) show rather similar climatological T,S values, roughly aligned along the same isopycnal ($27.25 kg/m^3$). PRED starts in this case far away from the HIST state (Figure 11c), although with similar

density conditions. The main difference with respect to the surface is that with the subsequent forecast years, the subsurface does not converge towards the typical mean HIST states. By the sixth forecast FMA the mean T,S value appears to stabilise along a weaker isopycnal ($27.1 kg/m^3$). This suggests that the forecast drift has brought the model to a different equilibrium state, at least in the Labrador Sea. The shock and the later drift may be caused by RECON being far from the EC-Earth3 model climate state in the Labrador Sea subsurface. In particular in terms of temperature and salinity (Figure 11), as the mean

density profiles are rather comparable due to the compensation between the temperature and salinity contributions (Figure 10). Similar T,S diagrams, using HIST as a baseline, will be used in the future when evaluating the suitability of different ocean reconstructions to initialise our next decadal prediction systems.

## 4 Summary and Conclusions

In this paper we have presented and evaluated the predictive skill of a decadal forecast system with EC-Earth, based on full-

field initialisation, that contributes to the Decadal Climate Prediction Project component A (DCPP-A). The main findings of the skill assessment are as follows:





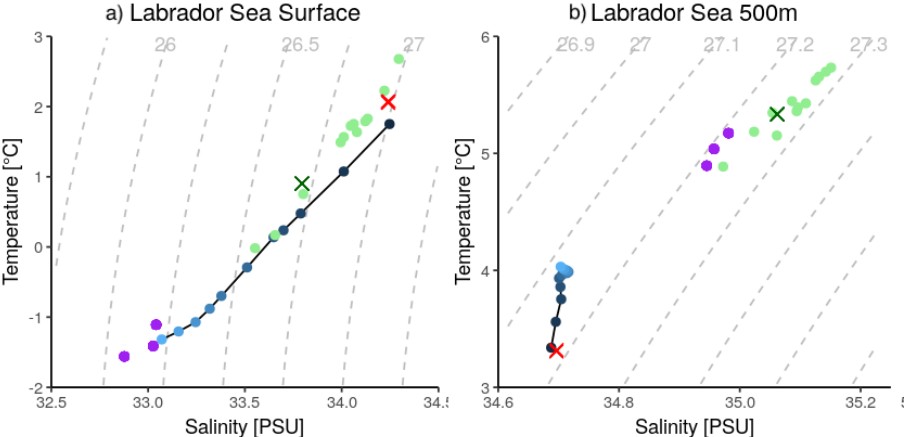

**Figure 11.** Scatterplot diagram between the climatological Labrador Sea temperature and salinity during the convection season (February-March-April) both a) at the surface and b) at 500m. Blue dots of different intensity represent the climatological PRED values as a function of forecast year, the red cross represents RECON and the green (purple) dots the HIST members with (without) active Labrador Sea convection. Isopycnals are represented by dashed grey lines in the background. The dark green line shows the HIST ensemble mean using all members.

– In agreement with other decadal forecast systems (e.g., Yeager et al., 2018; Robson et al., 2018), EC-Earth3 is able to skillfully simulate the global mean surface temperature at short (forecast year one) and long forecast times (forecast years 6-10), with a large part of the skill arising from changes in the external forcings.

– Comparing different skill metrics (i.e. anomaly correlation coefficient and mean square skill score; ACC and MSSS) in the predictions and in an ensemble of historical simulations we have shown a beneficial effect of initialisation. In the first forecast year, surface temperature anomalies in regions like the Tropical Pacific, the eastern Subpolar North Atlantic and the Southern ocean show added value from initialisation in the predictions. At longer forecast times only a few localised regions show improvements in terms of MSSS due to initialisation, exemplified by the Eastern Equatorial Pacific and the 480 Equatorial Atlantic. ACC differences show more limited improvements, from which we highlight a narrow band in the eastern Subpolar North Atlantic.

– The added value of initialisation is more easily discernible when considering both vertically and regionally integrated ocean quantities. For example, skill maps of the upper 300m ocean heat content (OHC300), which is more persistent than surface temperature as it is less affected by atmospheric perturbations, show larger areas of improved skill both 485 in the Pacific and Atlantic oceans. Likewise, skill metrics are systematically better in the initialised predictions for the Atlantic Multidecadal Varibility, although the improvements are not statistically significant.

– Another beneficial effect of initialisation is the reduction of the ensemble spread in the predictions with respect to the historical simulations, at least for the variables and indices analysed. The spread of the predicted anomalies is therefore better constrained at all forecast times.





– In contrast with other studies, the central Subpolar North Atlantic is a region of poor forecast skill in the EC-Earth3
forecast system. Both SST and the OHC300 show a detrimental effect of initialisation in the first 5 forecast years, which
could be explained by an initialisation shock.

To investigate this potential shock we have further explored the forecast evolution in a selection of key ocean variables
controlling multidecadal variability in the North Atlantic. The analysis showed that Labrador Sea convection collapses by
forecast year 3 in the predictions, leading to a rapid weakening of the Atlantic Meridional Overturning Circulation (AMOC)
and the Subpolar Gyre Circulation. This causes a cooling tendency of the western SPNA and a local expansion of sea ice,
which occupies the entire Labrador Sea by forecast year 10. Although a similar state of suppressed convection is found in 3 out
of 15 of the historical experiments, the mean of the historical ensemble (which represents the preferred model state towards
which the forecasts are expected to drift) exhibits higher AMOC and subpolar gyre strength values, regular convection in the
Labrador Sea and a more realistic sea ice extent. This suggests that the Labrador Sea convection collapse and subsequent North
Atlantic changes are associated with an initialisation shock that brings the predictions apart from their expected trajectory.

We have further related the Labrador Sea convection collapse to the evolution of local density stratification and the separate
contributions from temperature and salinity. During the first three forecast years, the Labrador density profile becomes more
strongly stratified than in most of the historical members with active convection, following an intense surface freshening. This
increase in stratification continues with forecast time, approaching but not reaching the strong density stratification levels from
the three historical members without convection. To assess if the forecasts actually drift to an attractor characterised by these
three historical members, we have additionally evaluated the climatological temperature and salinity in the region as a function
of forecast time at the surface and 500 m. At the surface, the predictions start with mean temperature and salinity conditions
within the range of those in the historical members with active convection, and by the end of the forecast they approach the
typical state of the members without convection. At the subsurface, however, the forecasts remain far from either of the typical
historical states, stabilising at forecast year 10 around a different (and lighter) attractor.

These results thus highlight the risk of initialising a sensitive region for decadal prediction, such as the Labrador Sea, too far
from its preferred model state. A problem that, in this case, could have been minimised by applying a weaker nudging in the
subsurface when producing the reconstruction that provided the ocean and sea ice initial conditions. Our findings also underline
the importance of reducing as much as possible the mean model biases in the Labrador Sea, in particular at the subsurface. The
problems herein described are particularly important when considering full-field initialisation (e.g., Magnusson et al., 2012;
Smith et al., 2013), an approach in which shocks of this kind are more prone to occur. In this sense, anomaly initialisation
emerges as a potential alternative to minimise the drifts and, more importantly here, to minimize the occurrence of initialisation
shocks. However, as previous studies have shown (e.g., Volpi et al., 2017), this approach is not exempt from problems, and
does not prevent initial model imbalances from happening, whose effect on Labrador Sea convection remains unknown. A
complementary alternative is to devote new efforts in climate model development to reduce the model biases over the region,
to thus reduce the mismatches with the observation-constrained products used for initialisation. Indeed, an appropriate model
tuning in the Labrador Sea would benefit decadal prediction in two ways. First, by improving in the model realism in a source
region of decadal skill, and the second, by helping to prevent or reduce problems associated with initialisation.



*Author contributions.* R.B. and S.W. led the analysis. R.B. and P.O. prepared the manuscript with contributions from all co-authors. M. D., P. O., Y. R.-R. and F.J. D.-R. contributed to the discussion and interpretation of the results. E. T., P. E. and M. C., made critical contributions to the development of the model version and workflow manager used to produce the experiments. J. A.-N., R. B., L.P.C. R. C.-G., V. S., E. T. and E. D. contributed to create the initial conditions for the decadal predictions. R. B. performed the decadal climate predictions and A. R., E. M.-C., I. C. the historical simulations. P.-A. B. and A. R. contributed to post-processing the model outputs. T. A., S. L.-T. and J. V.-R.
developed the code to compute key ocean diagnostics. A.-C. H. and N. P.-Z. developed the analysis tools.

*Competing interests.* The authors declare that they have no conflict of interest.

*Acknowledgements.* The work in this paper was supported by the European Commission H2020 projects EUCP (776613), APPLICATE (727862) and INTAROS (727890), a Spanish project funded by the Spanish Ministry of Economy (CLINSA, CGL2017-85791-R), Industry and Competitivity, a FRS-FNRS/FWO funded Belgian project (PARAMOUR, EOS-30454083) and an ESA contract (CMUG-CCI3-
TECHPROP). The climate simulations analysed in the paper were performed using the internal computing resources available at MareNostrum and addditional resources from PRACE (HiResNTCP, project 3: 2017174177) and the Red Española de Supercomputación (AECT-2019-2-0003 and AECT-2019-3-0006 projects) as well as the technical support provided by the Barcelona Supercomputing Center. In addition, several coauthors have been supported by personal grants: Y.R.-R, E.T. and .S. W. received funding from the European Union Horizon 2020 research and innovation programme (Grant Agreements 800154, 748750, and 754433, respectively); I.C. was supported by Generalitat
de Catalunya (Secretaria d'Universitats i Recerca del Departament d'Empresa i Coneixement) through the Beatriu de Pinós programme; J. A.-N. was supported by the Spanish Ministry of Science, Innovation and Universities through a Juan de la Cierva personal grant (FJCI-2017-34027); R. C.-G. was funded by the Spanish Ministry of Education, Culture and Sports with an FPU Grant (FPU15/01511); and M.D. and P.O. were funded by the Spanish Ministry of Economy, Industry and Competitivity through the Ramon y Cajal grants RYC-2017-22964 and RYC-2017-22772.





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
