# Peer review of "Assessment of a full-field initialised decadal climate prediction system with the CMIP6 version of EC-Earth"

_Earth System Dynamics, 2020_

## Referee Comment (RC1) · Panos Athanasiadis (Referee) · 5 Oct 2020

**REVIEW**

**Manuscript:** ESD-2020-66

**Title:** Assessment of a full-field initialised decadal climate prediction system with the CMIP6 version of EC-Earth

**Authors:** Roberto Bilbao, Simon Wild, Pablo Ortega, Juan Acosta-Navarro, Thomas Arsouze, Pierre-Antoine Bretonnière, Louis-Philippe Caron, Miguel Castrillo, Rubén Cruz-García, Ivana Cvijanovic, Francisco Javier Doblas-Reyes, Markus Donat, Emanuel Dutra, Pablo Echevarría, An-Chi Ho, Saskia Loosveldt-Tomas, Eduardo Moreno-Chamarro, Nuria Pérez-Zanon, Arthur Ramos, Yohan Ruprich-Robert, Valentina Sicardi, Etienne Tourigny, and Javier Vegas-Regidor.

**General Comments**

This study presents an assessment of the last BSC decadal hindcasts, run with the CMIP6 version of EC-Earth and contributing to DCPP-A. The manuscript is very well written. The study takes into account and discusses relevant aspects from the most recent literature in the field of decadal predictions. The analysis uses a variety of commonly used diagnostics and also goes in some depth into understanding (in the context of this model) the role of full-value initialization with the associated initialization shock and model drift occurring in a key area of deep-water formation in the North Atlantic. The Reviewer suggests this manuscript for publication after some minor amendments. The latter are mainly textual but include also the resizing of some figure panels and considering taking a look at surface wind biases (possibly to be shown as supplemental material).

**Specific Comments**

1. Line 5: It would be helpful to be admitted / clarified that the realistic initialization contains part of the externally forced trends as, for example, the oceans get warmer with global warming. Yes, there are also aerosols and $CO_2$ which modify radiation and clouds during the simulations, but the warming signal is also contained in the initialized ocean state (progressively warmer).

2. Line 6: "gets" → is

3. Line 13: "in the surface" → at the surface...... the subsurface layer,

4. Line 47-50: On this point, there is also another recent study using DCPP-A (Athanasiadis et al., 2020) that shows comparable (even higher) skill for the NAO using CESM-DPLE.

5. Line 61: "is initialisation" → is the realistic initialisation of the ocean state *(or of the Earth system, if the authors prefer).*

6.  Line 65: "especially in" → especially in the deep ocean and before modern instruments (such as ARGO floats) were introduced.

7.  Line 66: What is the meaning of the word "exclusively" in this sentence? Initial states are built from observations.

8.  Line 95: "that take" → which take

9.  Line 115: "10 member" → 10-member

10. Line 154: Has the word "cmorisation" been defined earlier? Perhaps it would be best to keep the "CMOR" part in capital letters.

11. Line 155: "data was…" → data were systematically checked for their quality with…

12. Line 174: I expect that the drift cannot affect equally all predictions (initialized in different years with different states, closer to or further from the model climatology). The drift is defined as the average tendency over many years, is not that so?

13. Line 199: "persisting it" → making it persist

14. Line 219: "Equator-60" → Equator–60 (not hyphen but en dash).

15. Line 248: What do you mean by "phases"? The Reviewer guesses what the authors might mean. Please take into account the common use of "phase" as a verb (https://www.merriam-webster.com/dictionary/phase) and expand this sentence accordingly.

16. Line 249: "equivalent" → comparable / similar

17. Line 259: "influence of" → influence of the unpredictable part of

18. Line 261: Here and elsewhere (where a similar expression is used as an adjective) use "low-frequency".

19. Line 295: "is for the some" → is for some

20. Line 319: "5f" is a reference to "Fig. 5f"? Please follow the instructions for authors of this journal – in any case, all references to figures should follow a standard way (same throughout the text).

21. Line 344: "ranges(Figure" (add space)

22. Line 351: "to aid" → so as to aid

23. Line 356: "feature" → behaviour / relationship

24. Line 359: "evolve" → evolves (singular)

25. Line 363:  Why should that be? Same model => same attractor.

26. Line 389:  The Reviewer questions the idea that PRED can reach (or come in the neighborhood of) the model's AMOC attractor in just 10 years.

27. Line 392:  But could not it be that a surface-wind bias (likely associated with a bias in Greenland blocking frequency) favors the formation of sea ice in the Lab. sea, which subsequently blocks heat and moisture surface fluxes? If the authors agree that this is a plausible explanation, at least in part, the Reviewer would suggest to take a look at surface wind biases in that area.

28. Line 414:  What does "their" refer to?

29. Line 421:  Here and elsewhere, please make "3" a superscript (exponent). Also, add some small spaces between values/numbers and units.

30. Line 428:  "simulation" → simulations

31. Line 486:  "varibility" → variability

32. Line 491:  Speaking of an "effect", is this positive, or negative? What kind of effect?

33. Line 498:  Speaking of different members exhibiting different mean states, it is likely that the AMOC has a degree of non-stationarity and not necessarily a uni-modal distribution. If that is so, then the multi-member time average may not correspond to any real attractor (preferred state).

34. Line 501:  "brings the predictions apart from" → carries the predictions away from

35. Line 517:  "prone" → likely
        A method is prone to errors, instead, the errors themselves are not "prone" to occur.

FIGURE 2:  In the caption please change the sentence referring to the hatching – what are significant are the ACC values, not the areas themselves. Also write: "Points with missing values…" as the masking is applied to an area.
From a scientific view point: the lack of predictive skill in the subpolar gyre (south of Greenland) is an indication of likely poor NAO skill (see Athanasiadis et al., 2020). Have the authors assessed the NAO skill for this set of hindcasts? If the NAO skill is indeed poor, perhaps it would be fair and worth mentioning this possible connection.

FIGURE 5:  This, but also other figures, should be expanded so as to best use the available space. The overall figure width should, however, remain a bit narrower than the width of the main text (plenty of space until there).
From a scientific view point: Why does HIST ensemble mean exhibit such a weak ENSO variability, in

contrast to PRED, for the 1$^{st}$ year of the predictions? Arguably, because ENSO events are mainly out of phase across the HIST ensemble (as expected). The authors may want to mention this rather trivial explanation.

FIGURE 11: "Scatterplot diagram" → Scatter plot.

---

## Referee Comment (RC2) · Stephen Yeager (Referee) · 27 Oct 2020

Review of "Assessment of a full-field initialised decadal climate prediction system with the CMIP6 version of EC-Earth", by Bilbao et al.

**General Comments**

This manuscript presents an overview assessment of the performance of the EC-Earth prediction system, contributed by BSC to CMIP6 DCPP. The skill assessment is focused on surface temperature (a focus that the authors might consider highlighting in the title), and includes assessment of modes of surface temperature variability and related ocean fields (heat content, AMOC, etc.). The second half of the study investigates the reasons for curiously low skill in the central subpolar North Atlantic and presents some interesting analyses that shed light on initialization shock and drift in this key region. The quality of the writing is very good and clear, with ample references to recent literature, and the quality of the figures is high (with some exception—see comment below). This study will be of interest to many in the decadal prediction community as it nicely documents the overall behavior of a single high-profile system (one of the WMO's Global Producing Centres for annual-to-decadal climate predictions). I therefore recommend publication after (mostly) minor revision.

**Specific Comments**

My specific recommendations for improvement:

1) Much of the paper elaborates on of the negative effects of an "initialisation shock" in the subpolar Atlantic, and this term is even included in the abstract. While the authors offer a definition of what this phrase means ("abrupt changes that occur soon after initialisation as a result of the adjustment of the climate model to the initial state"), I felt that the precise meaning of this term (and its usefulness for understanding system behavior) faded as I read. Certainly, there is a pathological adjustment to initialization going on in this system, but the distinction between shock and drift is not clear, nor is it clear that the initial shock (enhanced Labrador Sea convection) *causes* the longer term drift (towards reduced convection and sea ice expansion, AMOC decline, etc.). Is the shock really the essential problem in EC-Earth, or is it the drift towards ice-covered Labrador Sea? I suspect the latter is the more fundamental problem. I recommend a reconsideration of the phraseology used throughout.

2) Related to above, the skill improvement with lead time for NASPG-OHC300 (Figs. 5k and S1k) is interpreted as reflecting initialization shock behavior. However, the later figures (in particular, Fig. 7) make me question whether the relatively high skill for later lead times (e.g., LY7-10) is real skill. I note that HIST_NoConv exhibits a reasonably high correlation with RECON for Western SPNA-OHC300 (Fig. 7c) which is almost certainly spurious—it appears to relate to a post-1990s spinup of the NASPG in those members (Fig. 6b) which in turn appears related to a transition from fully ice-covered Lab Sea to only partially-covered Lab Sea, with associated increase in convection (Fig.

7). This mechanism for reproducing the late 20$^{th}$ century warming of the SPG is unequivocally unrealistic, even though it might yield higher correlation scores for NASPG-OHC300 than HIST itself (could you check this?). At long lead times, PRED seems to show similar behavior as HIST_NoConv (as noted in the text, but also in terms of Lab Sea transition from ice-covered_no-convection to partially-ice-covered_some convection), suggesting that the better NASPG-OHC300 "skill" at long lead times is a spurious artifact of an unrealistic warming mechanism. If true, this changes the interpretation of what is happening in the prediction system (i.e., it is not "initialization shock" followed by skill recovery via better representation of real mechanisms). If not true, how do the authors explain the increase in NASPG-OHC300 skill with lead time (Fig. 5k)?

3) Figures 8 and 10 have many small thin lines of various colors and hues that are very hard to distinguish (this reviewer is slightly color blind). Can a revised version be developed that is easier to see, particularly Fig. 8? I recognize that "easy to see" is quite subjective, and that these figures contain lots of information that is hard to display any other way. Perhaps the answer is "the figures are as clear as they can reasonably be" and I am in a small minority that has trouble viewing them, but if others (reviewers, coauthors, colleagues) also have difficulty with these figures then please make an effort to improve them.

Additional Comments (by line number)

63: This is not a complete sentence.
80: The meaning of "biases in the predictions" is not clear. Model mean bias is to be expected when using anomaly initialization. Do you mean "time-dependent biases in the predictions" (i.e. drift)?
111: ORCA has not been defined
124: There is no mention of how the land model component is initialized—can you please clarify?
205-207: Since sentence paragraphs are not advisable.
243: "signal" instead of "trend" to avoid awkward phrasing?
264: "associated with" instead of "to"
271: It would help to interpret Fig. 3 if the breakdown of MSSS into correlation and conditional bias terms were given explicitly (perhaps in section 2.3), and the corresponding relationships between Fig. 3 panels clarified (e.g., is panel a = panel d + panel g?).
286: Missing "(Figure"
302: There also seems to be noteworthy skill in the western tropical Pacific which should not be ignored.
315: I'm confused by this statement. Since both PRED and HIST show SER<1 in the first few months (Fig. 5c), aren't they both overconfident (under-dispersed)?
Fig. 5: It's unclear from the caption whether purple line (persistence forecast) is an ACC or MSSS score.
326, 340: It's not clear to me that the HIST spread is "excessive" and "too large" (although it is certainly larger than PRED) since I'm unsure how the concept of reliability

applies to uninitialized ensembles that aren't expected to be able to predict internal variability.

360: "black" should be "green"?

396: Fig. 7f is mislabelled as "e)"

Fig. 8: I find it very hard to make out the relevant details in this figure even after magnifying to 400%. Can you devise a better graphic that is more legible for color-challenged individuals? Same comment applies to Fig. 10. One simple option might be to just plot upper 400m to magnify the key region of interest. Another might be to plot as differences from HIST.

431: Please double check the sign of the restoring freshwater fluxes. Fig. 8 suggests that RECON is saltier at the surface than HIST (less stratified by salinity) which implies that a positive SALT flux (ie, negative freshwater flux) is used in the restoring.

451: Incorrect reference to figure 10 within this sentence.

Fig. 11: I think the last sentence of caption should be "dark green cross"?

501: Here and elsewhere, the distinction between "initialization shock" and model "drift" could be clarified. (also, what is the "expected trajectory"? a skillful one? one towards the model mean climatology?)

---

## Referee Comment (RC3) · G. A. Meehl (Referee) · 2 Nov 2020

General comments: This paper is a nice description of the characteristics of the EC-Earth initialized predictions for CMIP6 DCPP. The authors provide a welcome set of details regarding what they actually do and how they do it, something that other papers of this genre often do not do. They document features previously seen in other initialized prediction systems (e.g. better predictability in the Atlantic compared to the Pacific), and also note a difference in their system in that they have difficulties with predictions in the subpolar gyre region in the North Atlantic. They set about performing a detailed analysis to document why this difference is occurring in their model, and

uncover some interesting interrelated aspects involving overturning, stratification, bias and drift, and consequences for sea ice in that region. Overall, I think this paper makes an excellent contribution to ESD, and I have only minor comments.

Specific comments:

Line 42: Researchers at NCAR have documented the prediction of aspects of the IPO (e.g. Meehl et al. 2016) and have noted that the response to volcanic eruptions could explain in part why there is less overall predictability of the IPO compared to AMV (Meehl et al., 2015).

Line 83: A paper that should be referenced here that was important for documenting one of the main methods of bias adjustment that has subsequently been used is Doblas-Reyes, et al. 2013 (already in the reference list).

Lines 279-281: The authors need to explain more clearly what a negative value of MSSS means. They say in passing that PRED has lower ACC values than HIST, but more explanation would be helpful for the reader to interpret this important result which produces strikingly large areas of negative values in Fig. 3.

Lines 370-372: The authors note a very interesting feature in that their model drifts differently in two different periods. They should elaborate a bit more about this potentially very important aspect of their simulations that has profound implications for assessing prediction skill.

References

Meehl, G.A., H. Teng, N. Maher, and M.H. England, 2015: Effects of the Mt. Pinatubo eruption on decadal climate prediction skill. Geophys. Res. Lett., 42, 10,840-10,846, doi:10.1002/2015GL066608.

Meehl, G.A., A. Hu, and H. Teng, 2016: Initialized decadal prediction for transition to positive phase of the Interdecadal Pacific Oscillation. Nature Communications., 7, doi:10.1038/NCOMMS11718.

---

## Author Comment (AC1) · 3 Dec 2020

Specific Comments

1. Line 5: It would be helpful to be admitted / clarified that the realistic initialization contains part of the externally forced trends as, for example, the oceans get warmer with global warming. Yes, there are also aerosols and CO2 which modify radiation and clouds during the simulations, but the warming signal is also contained in the initialized ocean state (progressively warmer).

Reply: We agree with the reviewer's comment. Certainly the initial conditions include

a response to the external forcings, and can even correct part of the forced response that is misrepresented by the models and/or the forcings. To be more precise we have rewritten the sentence to simply say that most of the skill comes from the external radiative forcings.

2. Line 6: "gets" → is

Reply: corrected.

3. Line 13: "in the surface" → at the surface...... the subsurface layer,

Reply: corrected.

4. Line 47-50: On this point, there is also another recent study using DCPP-A (Athanasiadis et al., 2020) that shows comparable (even higher) skill for the NAO using CESM-DPLE.

Reply: The paragraph has been adjusted and the reference included.

5. Line 61: "is initialisation" → is the realistic initialisation of the ocean state (or of the Earth system, if the authors prefer).

Reply: suggestion accepted.

6. Line 65: "especially in" → especially in the deep ocean and before modern instruments (such as ARGO floats) were introduced.

Reply: suggestion accepted.

7. Line 66: What is the meaning of the word "exclusively" in this sentence? Initial states are built from observations.

Reply: The term "exclusively" has been removed.

8. Line 95: "that take" → which take

Reply: corrected.

9. Line 115: "10 member" → 10-member

Reply: corrected.

10. Line 154: Has the word "cmorisation" been defined earlier? Perhaps it would be best to keep the "CMOR" part in capital letters.

Reply: corrected.

11. Line 155: "data was..." → data were systematically checked for their quality with...

Reply: suggestion accepted.

12. Line 174: I expect that the drift cannot affect equally all predictions (initialized in different years with different states, closer to or further from the model climatology). The drift is defined as the average tendency over many years, is not that so?

Reply: The 'mean drift correction' that we apply assumes that drift is the same in all the forecasts, which may be a suitable approximation for certain variables. However, we agree with the reviewer's comment that drift is unlikely equal in all predictions, in fact, in section 3.3 we highlight how this is not the case for the AMOC and SPGSI. An inefficient drift removal may compromise the skill evaluation. In literature several drift correction methods have been proposed, but to date there is no clear advantage to using a particular method. We have rephrased the sentence to acknowledge that the underlying assumption (i.e. the insensitivity of the drift to the initial state) might not always hold.

13. Line 199: "persisting it" → making it persist

Reply: corrected.

14. Line 219: "Equator-60" → EquatorâŠ60 (not hyphen but en dash).

Reply: corrected.

15. Line 248: What do you mean by "phases"? The Reviewer guesses what the authors might mean. Please take into account the common use of "phase" as a verb (https://www.merriamwebster. com/dictionary/phase) and expand this sentence accordingly.

Reply: The verb has been changed to 'puts in phase'.

16. Line 249: "equivalent" → comparable / similar

Reply: corrected.

17. Line 259: "influence of" → influence of the unpredictable part of

Reply: corrected.

18. Line 261: Here and elsewhere (where a similar expression is used as an adjective) use "lowfrequency".

Reply: corrected.

19. Line 295: "is for the some" → is for some

Reply: corrected.

20. Line 319: "5f" is a reference to "Fig. 5f"? Please follow the instructions for authors of this journal – in any case, all references to figures should follow a standard way (same throughout the text).

Reply: corrected.

21. Line 344: "ranges(Figure" (add space)

Reply: corrected.

22. Line 351: "to aid" → so as to aid

Reply: corrected.

23. Line 356: "feature" → behaviour / relationship
Reply: corrected.

24. Line 359: "evolve" → evolves (singular)

Reply: corrected.

25. Line 363: Why should that be? Same model => same attractor.

Reply: What we mean to say is that the model might have more than 1 attractor, which seems to be the case given the existence of two different states of Labrador Convection. We have rephrased for clarity.

26. Line 389: The Reviewer questions the idea that PRED can reach (or come in the neighborhood of) the model's AMOC attractor in just 10 years.

Reply: We agree with the reviewer that ten years may be insufficient time to reach the model attractor(s), in particular for the AMOC. We have rephrased the sentence taking it into account.

27. Line 392: But could not it be that a surface-wind bias (likely associated with a bias in Greenland blocking frequency) favors the formation of sea ice in the Lab. sea, which subsequently blocks heat and moisture surface fluxes? If the authors agree that this is a plausible explanation, at least in part, the Reviewer would suggest to take a look at surface wind biases in that area.

Reply: To answer the reviewers comment we have looked at the surface wind stress over the Labrador Sea (See Supporting Figure 1). The plot shows that the drift in PRED is too small in comparison with the one in other variables and therefore seems unlikely that the wind is responsible for the very rapid sea ice growth.

Supporting Figure 1. Evolution of the FMA Windstress in the Labrador Sea in PRED for a) the meridional direction and b) zonal direction. Ensemble mean forecasts (10 members) of PRED are shown from blue to red every 3 startdates. Panels b) and d) are the climatological values as a function of forecast time.

28. Line 414: What does "their" refer to?

Reply: corrected.

29. Line 421: Here and elsewhere, please make "3" a superscript (exponent). Also, add some small spaces between values/numbers and units.

Reply: corrected.

30. Line 428: "simulation" → simulations

Reply: corrected.

31. Line 486: "varibility" → variability

Reply: corrected.

32. Line 491: Speaking of an "effect", is this positive, or negative? What kind of effect?

Reply: We refer to a negative effect on its regional skill. It's been rephrased to clarify it.

33. Line 498: Speaking of different members exhibiting different mean states, it is likely that the AMOC has a degree of non-stationarity and not necessarily a uni-modal distribution. If that is so, then the multi-member time average may not correspond to any real attractor (preferred state).

Reply: We agree with the reviewer's assessment. The fact that the historical ensemble mean might not represent a preferred state is now mentioned in the sentence.

34. Line 501: "brings the predictions apart from" → carries the predictions away from

Reply: suggestion accepted.

35. Line 517: "prone" → likely A method is prone to errors, instead, the errors themselves are not "prone" to occur.

Reply: corrected.

[Figure]

FIGURE 2: In the caption please change the sentence referring to the hatching – what are significant are the ACC values, not the areas themselves. Also write: "Points with missing values..." as the masking is applied to an area. From a scientific view point: the lack of predictive skill in the subpolar gyre (south of Greenland) is an indication of likely poor NAO skill (see Athanasiadis et al., 2020). Have the authors assessed the NAO skill for this set of hindcasts? If the NAO skill is indeed poor, perhaps it would be fair and worth mentioning this possible connection.

Reply: The caption of Figure 2 has been updated as suggested. We have looked at the NAO and we have low insignificant skill. The possible link between the low NAO skill and that of the SPNA OHC is now mentioned at the end of section 3.3.

FIGURE 5: This, but also other figures, should be expanded so as to best use the available space. The overall figure width should, however, remain a bit narrower than the width of the main text (plenty of space until there). From a scientific view point: Why does HIST ensemble mean exhibit such a weak ENSO variability, in contrast to PRED, for the 1st year of the predictions? Arguably, because ENSO events are mainly out of phase across the HIST ensemble (as expected). The authors may want to mention this rather trivial explanation.

Reply: The figures have been changed as suggested. The reviewer is correct, we have added a comment in the text.

FIGURE 11: "Scatterplot diagram" → Scatter plot.

Reply: corrected.

**a) tauvo**

**b) tauvo clim**

**c) tauuo**

**d) tauuo clim**

**Fig. 1.** Evolution of the FMA Windstress in the Labrador Sea in PRED for a) the meridional direction and b) zonal direction.

---

## Author Comment (AC2) · 3 Dec 2020

Specific comments:

Line 42: Researchers at NCAR have documented the prediction of aspects of the IPO (e.g. Meehl et al. 2016) and have noted that the response to volcanic eruptions could explain in part why there is less overall predictability of the IPO compared to AMV (Meehl et al., 2015).

Reply: These articles are now cited and have been added accordingly to the list of references.

[Figure]

Line 83: A paper that should be referenced here that was important for document-ing one of the main methods of bias adjustment that has subsequently been used is Doblas-Reyes, et al. 2013 (already in the reference list).

Reply: The paper is now cited also in this part of the manuscript.

Lines 279-281: The authors need to explain more clearly what a negative value of MSSS means. They say in passing that PRED has lower ACC values than HIST, but more explanation would be helpful for the reader to interpret this important result which produces strikingly large areas of negative values in Fig. 3.

Reply: A more detailed description of MSSS, what it represents, and the equations used to compute it has been included in the methods section (subsection 2.3). We have also changed Fig. 3 to make it more easily interpretable, and adjusted the corre-sponding discussion accordingly.

Lines 370-372: The authors note a very interesting feature in that their model drifts dif-ferently in two different periods. They should elaborate a bit more about this potentially very important aspect of their simulations that has profound implications for assessing prediction skill.

Reply: We have expanded the paragraph to discuss more at depth these non-stationary drifts and the need to address them with better drift correction techniques.

---

## Author Comment (AC3) · 3 Dec 2020

Review of "Assessment of a full-field initialised decadal climate prediction system with the CMIP6 version of EC-Earth", by Bilbao et al.

Reply to Dr. Steve Yeager:

Specific Comments:

My specific recommendations for improvement:

1) Much of the paper elaborates on the negative effects of an "initialisation shock" in the subpolar Atlantic, and this term is even included in the abstract. While the authors

offer a definition of what this phrase means ("abrupt changes that occur soon after initialisation as a result of the adjustment of the climate model to the initial state"), I felt that the precise meaning of this term (and its usefulness for understanding system behavior) faded as I read. Certainly, there is a pathological adjustment to initialization going on in this system, but the distinction between shock and drift is not clear, nor is it clear that the initial shock (enhanced Labrador Sea convection) causes the longer term drift (towards reduced convection and sea ice expansion, AMOC decline, etc.). Is the shock really the essential problem in EC-Earth, or is it the drift towards ice-covered Labrador Sea? I suspect the latter is the more fundamental problem. I recommend a reconsideration of the phraseology used throughout.

Reply: This is a really good point. We agree that both the initialization shock and the mean drift are closely related in our predictions, and that is not possible to disentangle from our analysis if the problem is caused by the processes behind the initial adjustment or by those related to the long-term drift, which might be related. We have tried to improve the clarity of their definitions in the introduction and explain how they might relate with one another. We also specify now that initial adjustments or shocks, when they occur systematically across start dates, can be regarded as the initial stage of the model drift, and even condition its later evolution. We have also carefully revised the rest of the paper to mention both the shock and the drift as the ultimate causes of the lack of skill in the central SPNA.

2) Related to above, the skill improvement with lead time for NASPG-OHC300 (Figs. 5k and S1k) is interpreted as reflecting initialization shock behavior. However, the later figures (in particular, Fig. 7) make me question whether the relatively high skill for later lead times (e.g., LY7-10) is real skill. I note that HIST_NoConv exhibits a reasonably high correlation with RECON for Western SPNA-OHC300 (Fig. 7c) which is almost certainly spurious—it appears to relate to a post-1990s spinup of the NASPG in those members (Fig. 6b) which in turn appears related to a transition from fully ice-covered Lab Sea to only partially-covered Lab Sea, with associated increase in

convection (Fig.7). This mechanism for reproducing the late 20th century warming of the SPG is unequivocally unrealistic, even though it might yield higher correlation scores for NASPG-OHC300 than HIST itself (could you check this?). At long lead times, PRED seems to show similar behavior as HIST_NoConv (as noted in the text, but also in terms of Lab Sea transition from ice-covered_no-convection to partially-ice-covered_some convection), suggesting that the better NASPG-OHC300 "skill" at long lead times is a spurious artifact of an unrealistic warming mechanism. If true, this changes the interpretation of what is happening in the prediction system (i.e., it is not "initialization shock" followed by skill recovery via better representation of real mechanisms). If not true, how do the authors explain the increase in NASPG-OHC300 skill with lead time (Fig. 5k)?

Reply: This is a really interesting hypothesis, which we had not thought of. To check if the hypothesis is true we have looked in more detail at the East and West SPNA OHC in the upper 300m (see Supporting Figure 1). As suggested, we have divided the historical ensemble into those members which exhibit convection (Hist_Conv, 7 out of 10 simulations) and those with suppressed convection (Hist_NoConv, 3 out of 10). The reviewer correctly pointed out that the historical members with no convection have higher ACC values than those with convection (Figure 1c), at least in the West-SPNA (note that to avoid differences in skill due to differences in the verification period we have set a common verification period: 1971-2008). This is because the members with no convection show a long-term warming trend which happens to coincide, in large part, with the observed one, even if it happens for the wrong reasons (that is the melting of Labrador Sea ice allowing for open ocean convection).

Even though PRED at the later forecast times reaches comparable ACC values to Hist_NoConv, these do not seem to be explained by the same mechanism explaining the spurious OHC300 warming in Hist_NoConv (Figure 1c). Indeed, the timeseries in Supporting Figure 1b shows that the large improvement in skill at the longest forecast range with respect to the first forecast years comes from a good representation of the
decadal variability, including the quick transition from cold to warm OHC anomalies that occurred during the mid-90s, with a radically different long-term evolution than in Hist_NoConv. The lack of skill in PRED at the initial forecast times comes from a poor representation of the observed inter-annual variability, which in the forecast shows some 'spikes' (or abrupt transitions) that might well result from the initialization shocks.

The respective plots for the eastern SPNA OHC300 can help explain the origin of the skill recovery at forecast years 7-10 over the whole SPNA. Indeed, they show that the eastern side of the region has rather constant predictive skill, comparable in PRED and both Historical ensembles, which implies that most of the skill might be forced. In terms of the mean gyre circulation, the eastern SPNA is upstream of the western side, and therefore the mean flow might be advecting the (skilfully) forecasted anomalies from the eastern into the western region, eventually substituting the unrealistic OHC anomalies generated in the first forecast years by the labrador convection collapse. If we take into account that the eastern SPNA maintains a similar level of skill all along the forecast, and that the western SPNA recovers it at the very end, we could then explain the increase in skill with lead time for the whole SPNA in Figure 5k. This is, of course, just one plausible hypothesis, but exploring it further would require extending the paper in a new direction, something that we would prefer not to do given that it is already lengthy and dense.

A reduced version of Supporting Figure 1 has been included in the Supplement, and is now discussed in the text to explain our hypothesis behind the whole SPNA skill recovery.

Supporting Figure 1: Timeseries and ACC skill of the West and East SPNA-OHC300 anomalies (with respect to 1971-2018). The timeseries and ACC have been computed for the common period for all forecast ranges (i.e. 1970-2018). The first two columns show the observed (grey bars) and predicted (PRED in red, HIST in blue, HIST_Conv in green and HIST_NoConv in purple) timeseries for the 1-4 and 7-10 forecast years respectively. The third column shows the ACC for PRED (red), HIST (blue), HIST_Conv

(green) and HIST_NoConv (purple). Statistically significant ACC values (at the 95% confidence level) are shown as empty circles.

3) Figures 8 and 10 have many small thin lines of various colors and hues that are very hard to distinguish (this reviewer is slightly color blind). Can a revised version be developed that is easier to see, particularly Fig. 8? I recognize that "easy to see" is quite subjective, and that these figures contain lots of information that is hard to display any other way. Perhaps the answer is "the figures are as clear as they can reasonably be" and I am in a small minority that has trouble viewing them, but if others (reviewers, coauthors, colleagues) also have difficulty with these figures then please make an effort to improve them.

Reply: The resolution and quality of the figures have been improved. In figures 8 and 10 the profiles now go down to 500m rather than 800m to allow for a better visualization of the near-surface differences. And in Figure 8 we now include two rows, one comparing PRED with RECON, and another comparing PRED with HIST, which reduces the amount of lines, and allows to see better the differences between RECON and PRED.

Additional Comments (by line number)

63: This is not a complete sentence.

Reply: corrected.

80: The meaning of "biases in the predictions" is not clear. Model mean bias is to be expected when using anomaly initialization. Do you mean "time-dependent biases in the predictions" (i.e. drift)?

Reply: We have rephrased it now as "skill degradation in the predictions".

111: ORCA has not been defined

Reply: Added information.

124: There is no mention of how the land model component is initialized—can you

please clarify?

Reply: Added information of the land model component (HTESSEL) and the initialisation.

205-207: Since sentence paragraphs are not advisable.

Reply: the sentence has been deleted as it was not necessary.

243: "signal" instead of "trend" to avoid awkward phrasing?

Reply: suggestion accepted.

264: "associated with" instead of "to"

Reply: corrected.

271: It would help to interpret Fig. 3 if the breakdown of MSSS into correlation and conditional bias terms were given explicitly (perhaps in section 2.3), and the corresponding relationships between Fig. 3 panels clarified (e.g., is panel a = panel d + panel g?).

Reply: The description of the MSSS in section 2.3 has been extended. We now include two equations, the one we used for the computation, that it's taken from Goddard et al. (2013; Eq. 5), and a more compact version that represents the numerator as the difference between two terms, one based on ACCs and another on conditional biases. Both equations are reproduced below. The re-arrangement in Equation 2 has allowed us to see that the middle and bottom plots in the former version of Figure 3 did not represent direct contributions to the MSSS. It is actually the differences in the squared ACCs/conditional biases that determine the final MSSS, and therefore those are the quantities that we now represent in the middle and bottom rows of Figure 3 to guide the interpretation of the MSSS. If we disregard the denominator (which is just a scaling factor with no impact on the sign to produce a skill metric that goes from -1 to 1), we can interpret the values in the upper row as the difference between the values on the second and the third row. The discussion on Figure 3 has been modified according to

the changes.

Also, because positive values in the difference between the squared ACC in PRED and the squared ACC in HIST do not necessarily correspond to a beneficial effect of initialization on skill (e.g. if the ACC in PRED is negative, and positive in HIST) we have decided to keep the plots on the differences in ACC from the former figure, which have been placed as a third row in Figure 2.

286: Missing "(Figure"

Reply: corrected.

302: There also seems to be noteworthy skill in the western tropical Pacific which should not be ignored.

Reply: added to the text.

315: I'm confused by this statement. Since both PRED and HIST show SER<1 in the first few months (Fig. 5c), aren't they both overconfident (under-dispersed)?

Reply: This has been corrected.

Fig. 5: It's unclear from the caption whether purple line (persistence forecast) is an ACC or MSSS score.

Reply: It has been indicated in the caption that the purple line refers to the persistence based on the ACC.

326, 340: It's not clear to me that the HIST spread is "excessive" and "too large" (although it is certainly larger than PRED) since I'm unsure how the concept of reliability applies to uninitialized ensembles that aren't expected to be able to predict internal variability.

Reply: The spread-error-ratio is a measure of reliability that has been typically applied both to initialised and non-initialised forecasts (Ho et al. 2013; Robson et al. 2018).

It evaluates if the typical distance between ensemble members is comparable to the typical distance between the individual members and the observations. Both terms are small at the beginning of an initialised forecast (because of initialization itself) and are expected to grow as the forecast progresses, although their ratio could vary, and converge to the one in the uninitialised experiments. That's why we show them both.

References:

Ho CK, Hawkins E, Shaffrey L, Bröcker J, Hermanson L, Murphy JM, Smith DM, Eade R (2013) Examining reliability of seasonal to decadal sea surface temperature forecasts: the role of ensemble dispersion. Geophys Res Lett 40(21):5770–5775

Robson, J., Polo, I., Hodson, D. L. R., Stevens, D. P., and Shaffrey, L. C.: Decadal prediction of the North Atlantic subpolar gyre in the HiGEM high-resolution climate model, Climate Dynamics, 50, 921–937, https://doi.org/10.1007/s00382-017-3649-2, 2018.

360: "black" should be "green"?

Reply: corrected.

396: Fig. 7f is mislabelled as "e)"

Reply: corrected.

Fig. 8: I find it very hard to make out the relevant details in this figure even after magnifying to 400%. Can you devise a better graphic that is more legible for color challenged individuals? Same comment applies to Fig. 10. One simple option might be to just plot upper 400m to magnify the key region of interest. Another might be to plot as differences from HIST.

Reply: To improve the visibility of the relevant details we have increased the size of the figure, zoomed it to the upper 500m, and duplicated the panels to compare separately PRED and RECON (for which the lines are closer to each other), and PRED and HIST.

431: Please double check the sign of the restoring freshwater fluxes. Fig. 8 suggests that RECON is saltier at the surface than HIST (less stratified by salinity) which implies that a positive SALT flux (ie, negative freshwater flux) is used in the restoring.

Reply: This has been corrected. The freshwater fluxes are defined in the model as going out of the ocean, while the heat fluxes into the ocean. The figure has been modified so both fluxes are into the ocean and the text adjusted.

451: Incorrect reference to figure 10 within this sentence.

Reply: corrected.

Fig. 11: I think the last sentence of caption should be "dark green cross"?

Reply: corrected.

501: Here and elsewhere, the distinction between "initialization shock" and model "drift" could be clarified. (also, what is the "expected trajectory"? a skillful one? one towards the model mean climatology?)

Reply: The sentence and all the other mentions to the drift and initialisation shock have been rewritten according to this comment and the first one.

[Figure]

**Fig. 1.** Timeseries and ACC skill of the West and East SPNA-OHC300 anomalies (with respect to 1971-2018).